# Cytokinin fluoroprobe reveals multiple sites of cytokinin perception at plasma membrane and endoplasmic reticulum

Karolina Kubiasová [1,8], Juan Carlos Montesinos[2,8], Olga Šamajová[3], Jaroslav Nisler [4,5], Václav Mik [4], Hana Semerádová[2], Lucie Plíhalová [4,5], Ondřej Novák [5], Peter Marhavý [2,6], Nicola Cavallari[2], David Zalabák[1], Karel Berka[7], Karel Doležal [4,5], Petr Galuszka[9], Jozef Šamaj [3], Miroslav Strnad[5], Eva Benková [2✉], Ondřej Plíhal [1,4,5✉] & Lukáš Spíchal [4✉]

Plant hormone cytokinins are perceived by a subfamily of sensor histidine kinases (HKs), which via a two-component phosphorelay cascade activate transcriptional responses in the nucleus. Subcellular localization of the receptors proposed the endoplasmic reticulum (ER) membrane as a principal cytokinin perception site, while study of cytokinin transport pointed to the plasma membrane (PM)-mediated cytokinin signalling. Here, by detailed monitoring of subcellular localizations of the fluorescently labelled natural cytokinin probe and the receptor ARABIDOPSIS HISTIDINE KINASE 4 (CRE1/AHK4) fused to GFP reporter, we show that pools of the ER-located cytokinin receptors can enter the secretory pathway and reach the PM in cells of the root apical meristem, and the cell plate of dividing meristematic cells. Brefeldin A (BFA) experiments revealed vesicular recycling of the receptor and its accumulation in BFA compartments. We provide a revised view on cytokinin signalling and the possibility of multiple sites of perception at PM and ER.

[1] Department of Molecular Biology, Centre of the Region Haná for Biotechnological and Agricultural Research, Faculty of Science, Palacký University, Šlechtitelů 27, 783 71 Olomouc, Czech Republic. [2] Institute of Science and Technology (IST), 3400 Klosterneuburg, Austria. [3] Department of Cell Biology, Centre of the Region Haná for Biotechnological and Agricultural Research, Faculty of Science, Palacký University, Šlechtitelů 27, 783 71 Olomouc, Czech Republic. [4] Department of Chemical Biology and Genetics, Centre of the Region Haná for Biotechnological and Agricultural Research, Faculty of Science, Palacký University, Šlechtitelů 27, 783 71 Olomouc, Czech Republic. [5] Laboratory of Growth Regulators, Institute of Experimental Botany of the Czech Academy of Sciences and Faculty of Science of Palacký University, Šlechtitelů 27, 783 71 Olomouc, Czech Republic. [6] Umeå Plant Science Centre, Department of Forest Genetics and Plant Physiology, Swedish University of Agricultural Sciences, 90183 Umeå, Sweden. [7] Department of Physical Chemistry, Regional Centre of Advanced Technologies and Materials, Faculty of Science, Palacký University, 17. listopadu 1192/12, 771 46 Olomouc, Czech Republic. [8] These authors contributed equally: Karolina Kubiasová, Juan Carlos Montesinos. [9] Deceased: Petr Galuszka. ✉email: eva.benkova@ist.ac.at; ondrej.plihal@upol.cz; lukas.spichal@upol.cz

The plant hormone cytokinin regulates various cell and developmental processes, including cell division and differentiation, embryogenesis, activity of shoot and root apical meristems, formation of shoot and root lateral organs and others[1]. Cytokinins are perceived by a subfamily of sensor histidine kinases (HKs), which via a two-component phosphorelay cascade activate transcriptional responses in the nucleus. Based on the subcellular localization of cytokinin receptors in various transient expression systems, such as leaf epidermal cells of tobacco (*Nicotiana benthamiana*), and membrane fractionation experiments of Arabidopsis and maize, the endoplasmic reticulum (ER) membrane has been proposed as a principal hormone perception site[2–4]. Intriguingly, recent study of the cytokinin transporter PURINE PERMEASE 14 (PUP14) has pointed out that the plasma membrane (PM)-mediated signalling might play an important role in the establishment of cytokinin response gradients in various plant organs[5]. However, localization of cytokinin HK receptors to the PM, although initially suggested[6], remains ambiguous. Here, by monitoring subcellular localizations of the fluorescently labelled cytokinin probe iP-NBD[7], derived from the natural bioactive cytokinin iP, and the cytokinin receptor ARABIDOPSIS HISTIDINE KINASE 4 (CRE1/AHK4) fused to GFP reporter, we show that pools of the ER-located cytokinin fluoroprobes and receptors can enter the secretory pathway and reach the PM. We demonstrate that in cells of the root apical meristem, CRE1/AHK4 localizes to the PM and the cell plate of dividing meristematic cells. Brefeldin A (BFA) experiments revealed vesicular recycling of the receptor and its accumulation in BFA compartments. Our results provide a revised view on cytokinin signalling and the possibility of multiple sites of perception at both PM and ER, which may determine specific outputs of cytokinin signalling.

## Results and discussion

**Cytokinin fluoroprobe iP-NBD shows affinity to receptors.** Fluorescently labelled analogues of phytohormones, including auxin, gibberellin, brassinosteroid and strigolactone, have been successfully used to map the intracellular fate of their receptors in planta[8]. To adopt this tool for mapping subcellular localization of cytokinin receptors, using docking experiments and cytokinin activity screening bioassays, we selected a fluorescently labelled bioactive compound that interacts with the binding site of a cytokinin receptor.

Cytokinin groups a collection of $N^6$-substituted adenine derivatives, including *trans*-zeatin (*t*Z) and isopentenyladenine (iP). They show different localization pattern and distinct partially overlapping functions in planta. *t*Z-type cytokinins play a role of acropetal messengers, whereas iP-type cytokinins operate as systemic or basipetal messengers[9]. The isoprenoid cytokinins (*t*Z- or iP-types) showed similar distribution patterns in different cell type populations within the root apex[10]. While *t*Z-type cytokinins were detected at much lower levels than other isoprenoid cytokinins, when concerns free cytokinin bases, the *t*Z content was found to be the highest among the free bases, followed by free iP that showed relatively enhanced content also in the stele[10]. Hence, iP seems to be a good candidate for a cytokinin fluoroprobe design. Moreover, iP is a natural cytokinin that cannot be transformed through *O*-glycosylation at the cytokinin side chain and thus the possibility of metabolic conversions of the cytokinin fluoroprobe by cytokinin deactivation enzymes in planta is minimized. Furthermore, covalent attachment of 7-nitro-2,1,3-benzoxadiazole (NBD), a small fluorophore, to the *N*9 position of iP eliminates a risk of a metabolic conversion of the final cytokinin fluorescent probe iP-NBD (Fig. 1a) through *N*-glycosylation, or formation of cytokinin

nucleotides. The stable attachment of the *N*9-substituent also prevents modifications at the *N*7 position by making this CK derivative completely inaccessible for *N*-glucosyltransferases[11]. Docking simulations using the CRE1/AHK4-iP crystal structure[12] and corresponding homology models suggested that iP-NBD may be fully embedded into the active sites of all AHK receptors (Fig. 1b) with micromolar range affinity. The affinity of iP-NBD to cytokinin receptors was measured using bacterially expressed recombinant AHK3 and CRE1/AHK4[13]. Both receptors share ligand preference for *t*Z, but AHK3 has about tenfold lower affinity towards iP compared to CRE1/AHK4[13,14]. Competitive binding assays with *E. coli* expressing either AHK3 or CRE1/AHK4[15] showed that iP-NBD competes for receptor binding with radiolabelled natural cytokinins iP and *t*Z in different ranges of ligand concentrations (Fig. 1c; Supplementary Fig. 1a), corresponding with the receptor ligand preferences. As predicted, iP-NBD had lower affinity to AHK3 (with $K_i \sim 37\,\mu M$ and $>100\,\mu M$ against radiolabelled *t*Z and iP, respectively) than to CRE1/AHK4 (with $K_i \sim 1.4\,\mu M$ and $\sim 31\,\mu M$ against radiolabelled *t*Z and iP, respectively), indicating that this fluoroprobe is more specific to CRE1/AHK4 (Fig. 1c; Supplementary Fig. 1a). Docking into the CRE1/AHK4-iP crystal structure[12] showed that iP-NBD binds into the receptor cavity in a similar manner to iP, but the lack of interaction via *N*9 (which links the fluorescent probe) causes the purine ring shift leading to the larger distance and thus weaker interaction between *N*7 and Asp137 (Fig. 1b). Despite iP-NBD being accommodated into the cytokinin-binding pockets of the receptors, it showed limited ability to trigger cytokinin response in *E. coli* ($\Delta rcsC$, *cps::lacZ*) receptor activation assay[13] (Supplementary Fig. 1b). In Arabidopsis seedlings, iP-NBD in a concentration-dependent manner significantly increased the expression of the early cytokinin response gene *ARABIDOPSIS RESPONSE REGULATOR5* (*ARR5*) already 15 min after its application, suggesting that the synthetic cytokinin fluoroprobe can activate cytokinin signalling pathway in planta (Fig. 1d; Supplementary Fig. 1c). In comparison to iP, a natural cytokinin, iP-NBD triggered cytokinin response with significantly lower efficacy and when applied together with iP no additive effect on the *ARR5* expression could be detected (Fig. 1d). In the *pTCSn::ntdTomato:TNOS* cytokinin reporter assay[16], iP-NBD did not increase expression of the reporter 6 h after treatment, but when applied simultaneously with iP, iP-NBD partially attenuated iP-mediated enhancement of the TCS reporter expression (Supplementary Fig. 1d). Altogether, these analyses suggest partial agonistic mode of action of iP-NBD that binds to a cytokinin receptor and activates it with only minimal efficacy compared to a natural cytokinin ligand. At excess concentrations, iP-NBD is then acting as a competitive antagonist, competing with the full agonist (a natural cytokinin) for receptor occupancy. Altogether, the above experiments show that iP-NBD binds to cytokinin receptors and has potential for specifically tracking their subcellular localization in planta.

**Biological characteristics of iP-NBD.** To reliably monitor iP-NBD distribution in planta, we first evaluated its biological stability, fluorescence characteristics and saturation kinetics. iP-NBD stability across the different pH conditions that appear in apoplast, cytosol and different cell organelles was tested in vitro in the pH ranging from 4 to 8 by quantitative liquid chromatography-tandem mass spectrometry (LC-MS/MS). No significant changes of iP-NBD concentration were found in the buffered solutions under both 6 and 16 h of incubation pointing to a broad pH stability of iP-NBD (Supplementary Fig. 2a). Taking into account the chemical structure of iP-NBD that prevents *O*- and/or *N*-glycosylation, the presumed in planta catabolic pathway of this

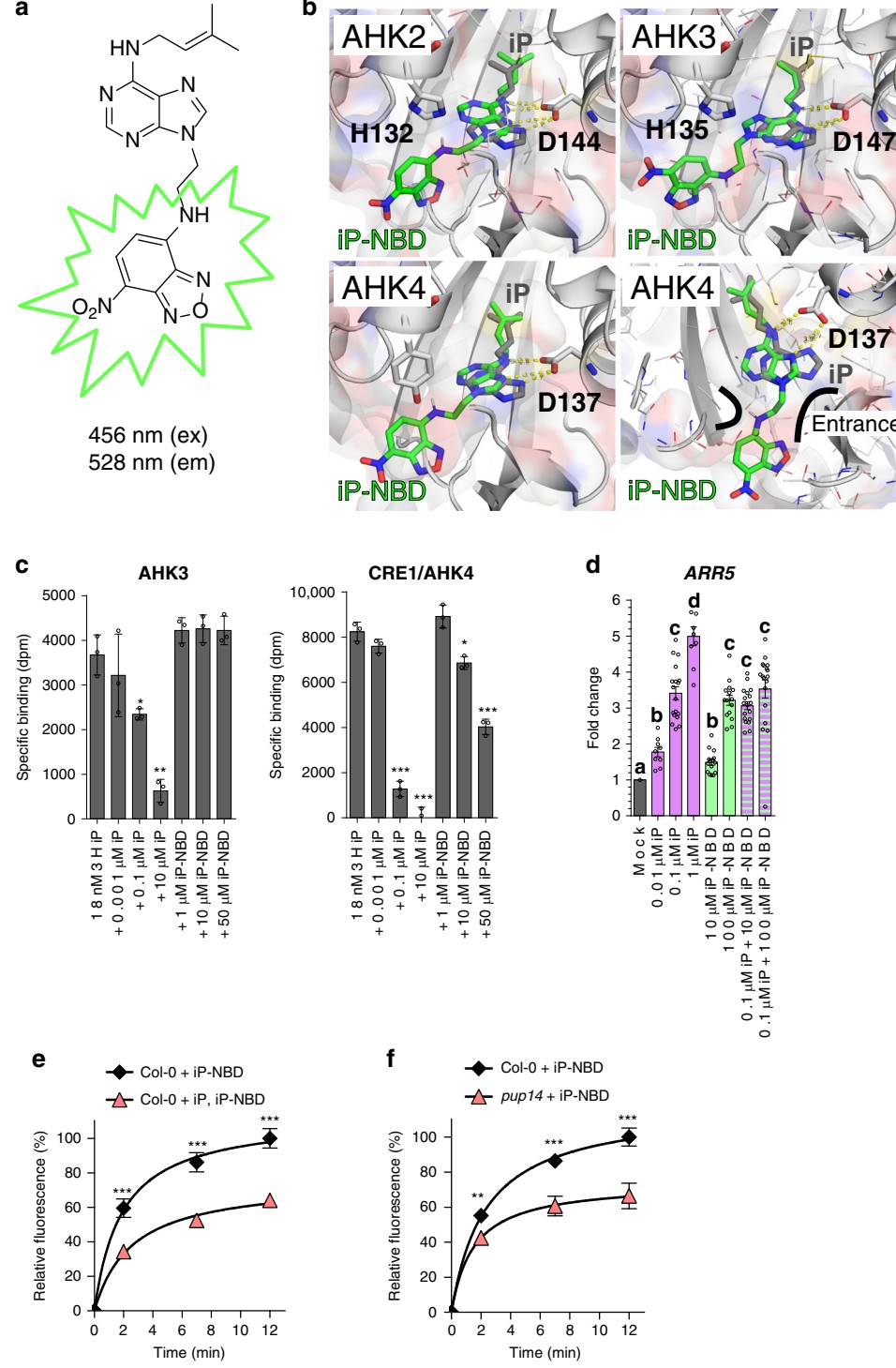

**Fig. 1 iP-NBD interacts with cytokinin perception and transport. a** Chemical structure of iP-NBD with excitation and emission wavelengths **b** Superposition of docking simulation of iP-NBD (in green) and natural ligand iP (in grey) in AHK2, AHK3 (upper row) and CRE1/AHK4 (left panel of lower row) receptor cavity showing the embedded position of the ligands. CRE1/AHK4 (right panel of lower row) with visualized entrance channel and the fluorophore fitting into an antechamber through the ethylene linker. **c** Competitive binding assay with *Escherichia coli* expressing AHK3 and CRE1/AHK4. Binding of [2-³H]iP (18 and 6 nM in the case of AHK3 and CRE1/AHK4, respectively) was assayed together with increasing concentrations of unlabelled iP and iP-NBD. The bars represent mean ± s.d., \***p < 0.001, \*\*p < 0.01, \*p < 0.05; n = 3 (Student's t test). **d** Expression of the early cytokinin response gene *ARR5* in 5-day-old seedlings of Col-0 treated with 0.01 μM iP, 0.1 μM iP, 1 μM iP, 10 μM iP-NBD, 100 μM iP-NBD or co-treatments of 0.1 μM iP + 10 μM iP-NBD, 0.1 μM iP + 100 μM iP-NBD or DMSO for 15 min (mean ± s.d., p < 0.05 by two-way ANOVA. n = 9–18; three technical replicates from three biological replicates per condition). **e** Kinetics of 5 μM iP-NBD uptake in Arabidopsis LRC cells of wild type (Col-0) pre-treated with 5 μM iP. **f** Kinetics of 5 μM iP-NBD uptake in root meristem epidermal cells of wild type (Col-0) and *pup14*. iP-NBD fluorescence was measured in four time points (0, 2, 7 and 12 min). The bars represent mean ± s.d., \***p < 0.001, \*\*p < 0.01; n ≥ 20 (Student's t test) (**e, f**).

molecule might be $N^6$ side-chain cleavage by endogenous CYTOKININ OXIDASE/DEHYDROGENASE (CKX) activity. Hence, the in vivo stability of the fluoroprobe was tested. iP-NBD was applied to Arabidopsis cells, and its intracellular processing was followed over a period of 0.5–5 h by LC-MS/MS analysis. Thus, iP-NBD and $N9$-NBD-labelled adenine (Ade-NBD), the expected product of iP-NBD deprenylation by CKXs, were used as molecular standards. Under these conditions, iP-NBD showed high stability within the first 30 min (≥90% recovery of intact molecule), dropping drastically after 5 h. The concentration of Ade-NBD steadily increased, reaching the maximal concentration after 4 h (Supplementary Fig. 2b). The fact that iP-NBD can be recognized by CKXs as a substrate was confirmed by in vitro enzymatic reaction with AtCKX2, one of the most active CKX isoforms with an apoplastic localization[17]. AtCKX2 converted iP-NBD to the product with approx. six times lower turnover rate $k_{cat}$ compared to the parental iP molecule, but only with 33% lower catalytic efficiency $V_{max}/K_m$ (Supplementary Fig. 2c).

**Internalization of iP-NBD follows rapid saturation kinetics.** In terms of fluorescent characteristics, the emission maximum of the cytokinin fluoroprobe was in the yellow-green part of the spectrum at 528 nm suitable for co-localization with fluorescent markers emitting at red wavelengths (Supplementary Fig. 2d, f). Quantitative fluorescence microscopy of wild-type plants (Col-0) showed that cellular internalization of iP-NBD followed rapid saturation kinetics, reaching a plateau after approximately 12 min (Fig. 1e). Pre-treatment with non-labelled iP and subsequent application of iP-NBD resulted in a significant reduction of intracellular iP-NBD fluorescence (Fig. 1e). This suggested that transport and/or intracellular binding competition between iP-NBD and the natural cytokinin competitor was taking place, further pointing to the cytokinin-like properties of the iP-NBD molecule. Significantly slower progression of iP-NBD accumulation in cells of a *pup14* mutant (lacking the functional cytokinin transporter PUP14) confirmed that specific cytokinin transport partially accounts for the amount of iP-NBD detected intracellularly (Fig. 1f). Unlike iP-NBD, Ade-NBD, which lacks the cytokinin-specific side chain, has no affinity to the cytokinin receptors (Supplementary Fig. 2e) and exhibited a weak diffused apoplastic and patchy intracellular signal in epidermal cells (Supplementary Fig. 2f).

**iP-NBD co-localizes with ER, TGN and early endosomal markers.** Affinity of iP-NBD to cytokinin receptors, in particular to CRE1/AHK4, motivated us to monitor subcellular localization of this cytokinin fluoroprobe, aiming to trace potential sites of interaction with the receptor. Two cell types, namely differentiated lateral root cap (LRC) cells and epidermal cells at the root meristematic zone of Arabidopsis root, were selected for in-depth analyses. In a line with reported ER-localization of the AHK cytokinin receptors[2,3], iP-NBD co-localized with p24δ5-RFP, an ER-specific marker, in both cell types (Fig. 2a, b, red arrowheads; Supplementary Table 1). Notably, we also detected strong iP-NBD fluorescence signal in distinct spot-like structures, which did not overlap with the ER reporter (Fig. 2a, b; white arrowheads). Likewise, co-visualization with HDEL-RFP, an ER-specific marker, corroborated dual ER and spot-like localization of iP-NBD in both LRC and epidermal cells (Supplementary Fig. 3a, b).

To further explore the nature of peripheral and spot-like subcellular structures showing affinity to iP-NBD, we performed co-staining with FM4-64, the membrane selective dye labelling PM and endosomal/recycling vesicles in plant cells[18]. In both epidermal and LRC cells, iP-NBD signal was detected intracellularly and partially co-localized with the FM4-64 stained vesicles corresponding to internalized and recycling endosomes (Fig. 2c, e; Supplementary Fig. 3c). Interestingly, detailed profiles of fluorescence intensity distributions of iP-NBD and FM4-64 revealed their partial co-localization at the PM of epidermal cells, which was not the case for LRC (Fig. 2d compared to Fig. 2f). These observations suggested that apart from ER, iP-NBD might accumulate in subcellular vesicles and at the PM.

To gain further insights into iP-NBD subcellular localization and to test its affinity to endomembrane structures, we analysed the impact of brefeldin A (BFA), a fungal toxin, inhibiting ER-Golgi and post-Golgi trafficking to the PM and to vacuoles, thus causing formation of endosomal clusters, so-called BFA compartments[19]. Strikingly, in root epidermal cells, we observed accumulation of iP-NBD signal in clusters corresponding to BFA compartments stained with FM4-64 (Fig. 2g, blue arrowheads). Co-localization with RabA1e-mCherry, a BFA-sensitive endosome/recycling endosome marker, provided additional supporting evidence that in root epidermal cells iP-NBD exhibits affinity to vesicular endomembrane system where subpopulations of cytokinin receptors may be localized (Supplementary Fig. 3d; Supplementary Table 1). Next, we traced the localization of the cytokinin fluoroprobe using a set of Wave marker lines specific for various subcellular organelles[20]. Notably, in root epidermal cells, we observed a partial co-localization of iP-NBD with a *cis*-Golgi (GA) marker, SYP32-mCherry (Supplementary Fig. 3e; Supplementary Table 1), an integral GA membrane protein, Got1p homologue-mCherry (Supplementary Fig. 3f; Supplementary Table 1), and with TGN/early endosome marker, VTI12-mCherry (Supplementary Fig. 3g; Supplementary Table 1). Interestingly, iP-NBD did not co-localize with a late endosome marker, RabF2b/W2R-mCherry (Supplementary Fig. 3h; Supplementary Table 1) nor with a vacuolar marker, VAMP711-mCherry (Supplementary Fig. 3i; Supplementary Table 1). In cells of LRC, we observed partial co-localization with the GA markers, SYP32-mCherry (Supplementary Fig. 4a; Supplementary Table 1) and Got1p homologue-mCherry (Supplementary Fig. 4b; Supplementary Table 1), an endosome/recycling endosome marker RabA1e-mCherry (Supplementary Fig. 4c; Supplementary Table 1) and with the TGN/early endosomal marker VTI12 (Supplementary Fig. 4d; Supplementary Table 1). However, no co-localization was detected with late endosomal RabF2b-mCherry (Supplementary Fig. 4e; Supplementary Table 1) or vacuolar VAMP711-mCherry markers (Supplementary Fig. 4f; Supplementary Table 1).

Overall, monitoring of iP-NBD in LRC and epidermal cells corroborate the ER as an organelle with affinity to cytokinin. However, co-localization of iP-NBD with TGN and early endosomal markers as well as its accumulation in BFA compartments indicate that proteins with affinity to iP-NBD, such as cytokinin receptors, do not reside exclusively at ER, but may enter the endomembrane trafficking system and possibly localize also to the PM.

**Generation and isolation of CRE1/AHK4-GFP transgenic lines.** Previously, ER-localization of Arabidopsis cytokinin receptors has been demonstrated using transiently transformed *Nicotiana benthamiana*[2,3] and Arabidopsis[3], and by employing cytokinin-binding assays with fractionated Arabidopsis cells expressing Myc-tagged receptors[2]. However, so far no experimental support has been provided for their possible entry into the subcellular vesicular trafficking and PM localization. Yet, the possibility of cytokinin HKs localization to the PM has been hypothesized within a context of an integrative model for cytokinin perception and signalling[21]. The potential sites of CKs perception had been questioned in relation to the pH dependence of the binding by HKs. It was

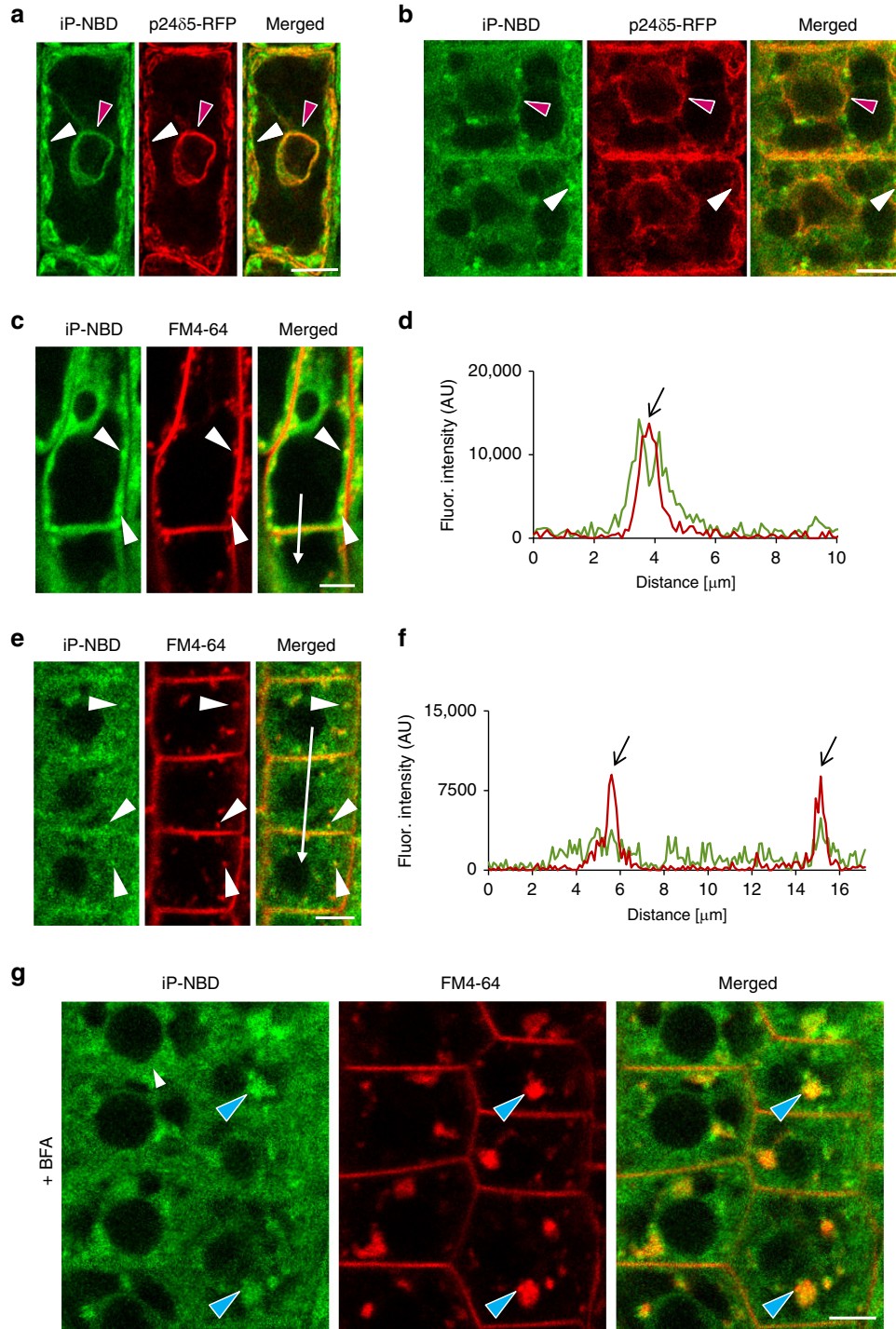

**Fig. 2 Monitoring of fluorescently labelled cytokinin iP-NBD in cells of Arabidopsis root. a, b** Monitoring of fluorescently labelled cytokinin iP-NBD (green) and ER-marker p24δ5-RFP (red) in LRC cells (**a**) and root meristematic epidermal cells (**b**). iP-NBD detected partially co-localizing with p24δ5-RFP in ER (red arrowheads) and in non-ER cellular structures (white arrowheads). **c–f** Monitoring of iP-NBD (green) and FM4-64 (red, membrane selective dye) in LRC (**c, d**) and root meristematic epidermal cells (**e, f**). White arrowheads (**c, e**) indicate co-localization of iP-NBD and FM4-64 in vesicles. Profiles of fluorescence intensity distribution of both FM4-64 (red line) and iP-NBD (green line) in LRC (**d**) and epidermal (**f**) cells were measured along the white lines (**c, d**) starting from upper end (0 μm) towards the arrowhead. Peaks of FM4-64 fluorescence maxima (black arrows) correlate with the plasma membrane staining. iP-NBD fluorescence maximum does not overlap with FM4-64 fluorescence peak at the plasma membrane in LRC cell (**d**). Peaks of iP-NBD signal partially overlap with FM4-64 maxima and indicate presence of cytokinin fluoroprobe at the plasma membrane of epidermal cells (**f**). **g** Co-localization of iP-NBD and FM4-64 in endosomal compartments (blue arrowheads) formed in root meristematic epidermal cells treated with 50 μM in BFA for 1 h. Scale bars = 5 μm.

shown that CK binding to AHK3 is pH dependent with optimum at basic pH and a dramatic decrease at acidic pH[14]. This finding fits with the ER-localization of CK receptors and has also been used to cast doubt on PM-function of the receptors due to the acidic pH of apoplast acting as a constraint on efficient CK binding. However, in contrary to AHK3, CRE1/AHK4 affinity was shown not to be dramatically altered at acidic pH[14]. Importantly, a recent work by Jaworek et al.[22] shows detailed analysis of pH influence on binding strength of CKs to the receptors from poplar (Populus × canadensis cv. Robusta). They showed that CK binding to PcHK3 (ortholog of AHK3) steadily increases towards higher pH values, whereas binding to PcHK4 (ortholog of CRE1/AHK4) linearly decreased from an optimum for ligand binding at pH 5.5. These findings support the idea that CRE1/AHK4 can effectively sense CKs from the apoplast.

The only cytokinin receptor studied for its localization using stably transformed Arabidopsis plants was AHK3 [3]. Unlike this receptor, subcellular localization of CRE1/AHK4 has not been addressed in much detail. Taking into account a higher affinity of iP-NBD to this receptor, we focused on monitoring its subcellular localization using Arabidopsis stable transgenic lines carrying CRE1/AHK4-GFP construct driven by a constitutive 35S promoter. Two independent lines displaying significantly increased transcription of CRE1/AHK4-GFP when compared to wild type were selected for detailed observations (Supplementary Fig. 5a). Western blot analyses confirmed accumulation of the CRE1/AHK4-GFP product of proper ~150 kDa size in both lines, although lower levels of the fusion protein were detected in the 35S::CRE1/AHK4-GFP line (1) when compared to the line (2) (Supplementary Fig. 5b, c). To test the functionality of the CRE1/AHK4-GFP fusion protein, we performed transient expression assays in Arabidopsis protoplasts. Co-expression of 35S::CRE1/AHK4-GFP with a cytokinin sensitive reporter TCS::LUCIFERASE (TCS::LUC) resulted in 85 ± 6.9-fold upregulation of the reporter activity by cytokinin when compared to protoplasts co-transformed with controls (plasmids carrying either GFP or GUS reporter only resulting in 28 ± 2.4- and 32 ± 1.5-fold increase of LUCIFERASE activity, respectively) (Supplementary Fig. 5d). In planta, functionality of the CRE1/AHK4-GFP was tested by expression analyses of the type-A early cytokinin response genes in the 35S::CRE1/AHK4-GFP transgenic lines. Application of cytokinin resulted in strong upregulation of ARR5 and ARR7 in wild type and both transgenic lines expressing CRE1/AHK4-GFP (Supplementary Fig. 5e). However, a significantly enhanced transcription of ARR5 and ARR7 in response to cytokinin compared to wild type was detected only in 35S::CRE1/AHK4-GFP line (2), which displayed a higher accumulation of CRE1/AHK4-GFP. ARR5 and ARR7 have been reported as being among the most sensitive type-A early cytokinin response genes, reaching expression maxima within 10–15 min following cytokinin application[23]. We argued that a high responsiveness of these genes to cytokinin might hinder detection of more subtle changes in cytokinin sensitivity in line with lower expression of the CRE1/AHK4-GFP. When compared to ARR5 and ARR7, ARR16 showed maximum transcription within 40–60 min following cytokinin application[23]. A significantly higher expression of ARR16 after cytokinin application for 15 min was detected in both CRE1/AHK4-GFP overexpressing lines when compared to wild type. These results suggest that proportionally with levels of CRE1/AHK4-GFP expression, the sensitivity of both lines to cytokinin stimulus is enhanced (Supplementary Fig. 5e), indicating that CRE1/AHK4-GFP maintains its biological activity.

Transgenic Arabidopsis lines expressing CRE1/AHK4-GFP exhibited phenotypes typical of plants with enhanced activity of cytokinin such as a shorter primary root, slower root growth rate and decreased lateral root density (Supplementary Fig. 5f–i). Both transgenic lines expressing CRE1/AHK4-GFP displayed hypersensitive-like responses to exogenous cytokinin treatment on the primary root growth compared to the wild-type control, and in contrast to cytokinin insensitive ahk4/cre1-2 loss-of-function mutant (Supplementary Fig. 5j).

**CRE1/AHK4-GFP co-localizes with the ER and the PM markers.** As previously reported and in line with iP-NBD subcellular localization, CRE1/AHK4-GFP in LRC and epidermal cells of root apical meristem co-localized with ER marker p24δ5-RFP (Fig. 3a–c, red arrowheads). Intriguingly, in epidermal cells of the root meristematic zone, CRE1/AHK4-GFP signal at the PM area, not co-localizing with ER reporter, could also be detected (Fig. 3d, e). Subsequent analysis revealed strong overlap of CRE1/AHK4-GFP with the PM reporter PIP1;4-mCherry and NPSN12-mCherry (Fig. 4a–d), thus hinting at localization of the cytokinin receptor at the PM. Moreover, in dividing meristematic cells CRE1/AHK4-GFP could also be detected at the expanding cell plate (Fig. 4c–f, asterisks) while it co-localized there with established cell plate vesicular marker FM4-64 (Fig. 4e, f). Importantly, it has been shown that during cytokinesis the cell plate might receive material both from post-Golgi compartments as well as from the PM through sorting and recycling endosomes[24]. Hence, detection of CRE1/AHK4-GFP at the cell plate provides further supporting evidence that the cytokinin receptor might reside outside of ER, namely on cytokinetic vesicles forming cell plate[25]. Further evidence confirming localization of CRE1/AHK4-GFP to the PM resulted from the subcellular study using super-resolution structural illumination microscopy (SIM)[26]. This SIM analysis revealed co-localization of CRE1/AHK4-GFP with FM4-64 labelled PM with average Pearson's coefficient 0.345 ± 0.113 (n = 30; Fig. 4g; Supplementary Fig. 6a). Unlike epidermal cells of the root meristematic zone, in LRC cells the CRE1/AHK4-GFP signal resided in the ER and no co-localization with a PM reporter (NPSN12-mCherry) could be detected (Supplementary Fig. 6b, c). Inhibition of endocytic trafficking and vesicular recycling in meristematic cells by BFA resulted in co-accumulation of CRE1/AHK4-GFP and FM4-64 in the BFA compartments in line with the presence of the receptor in the endomembrane system (Fig. 4h). Wash-out of BFA allowed re-localization of the cytokinin receptor back to the PM indicating that it might cycle between PM and TGN (Supplementary Fig. 6d). Although occasionally in some cells of LRC co-staining with FM4-64 revealed CRE1/AHK4-GFP in BFA compartments, they were relatively rare and randomly scattered in some LRC cells indicating that CRE1/AHK4-GFP trafficking in differentiated cells of LRC might differ from that observed in epidermal cells of root apical meristem (Supplementary Fig. 6e). Importantly, no accumulation of the ER marker p24δ5-RFP in the BFA compartments in either epidermal cells of meristem (Fig. 4i) or LRC cells (Supplementary Fig. 6f) could be detected, suggesting that CRE1/AHK4-GFP signal is specifically enriched in the BFA bodies and not related to structural changes of ER in BFA-treated cells.

Altogether, these results indicate that in LRC cells CRE1/AHK4 may reside preferentially at the ER, whereas in epidermal cells of the root apical meristem the cytokinin receptor can enter the endomembrane system and localizes both at the ER and at the PM.

To further explore whether cytokinin receptor might occupy different subcellular location in cells at distinct stage of differentiation, we monitored CRE1/AHK4-GFP in different cell types. Similarly to epidermis, in provasculature cells in the root meristematic zone, the CRE1/AHK4-GFP seems to localize at the ER, the PM and at the cell plate of dividing stele cells (Supplementary Fig. 6g). To strengthen the conclusion that in meristematically active cells cytokinin receptor might enter the secretory pathway and reach the PM, we performed

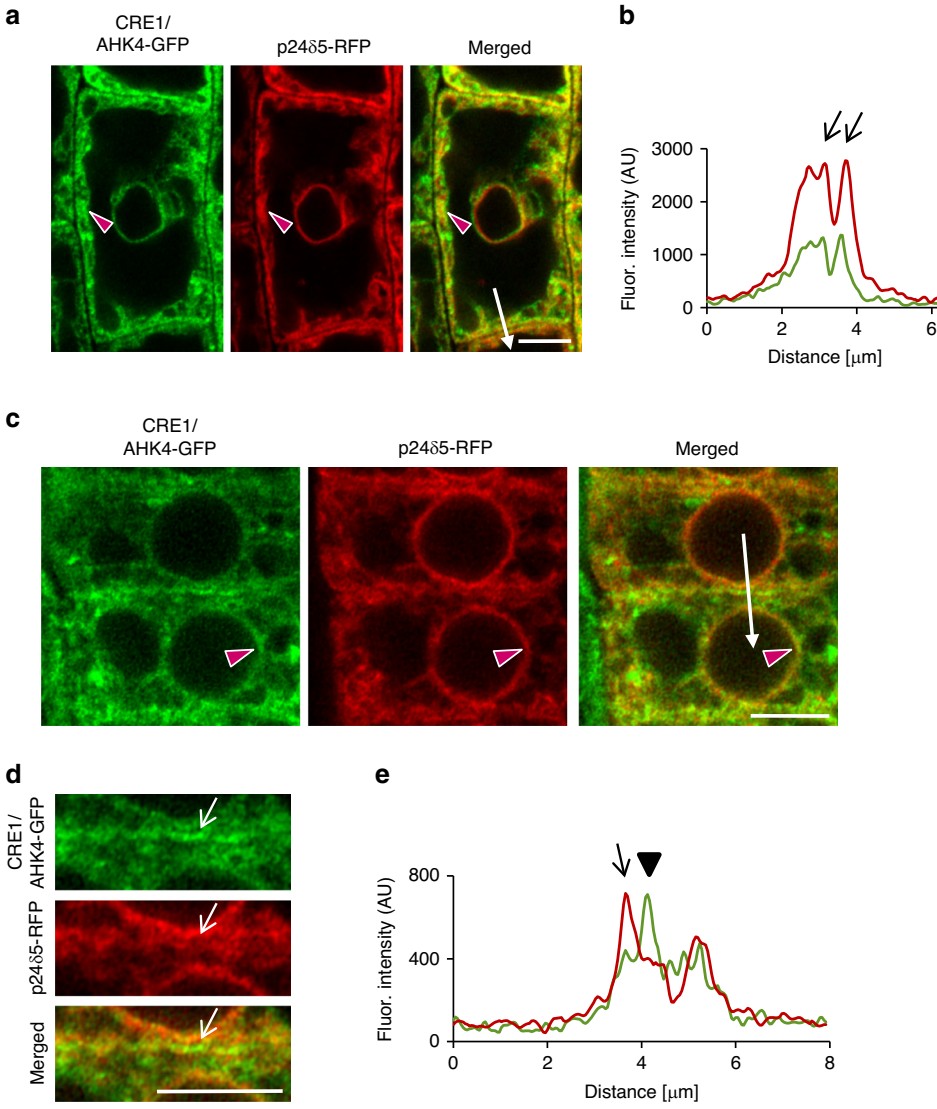

**Fig. 3 CRE1/AHK4-GFP subcellular localization in cells of Arabidopsis root. a–e** Monitoring of CRE1/AHK4-GFP cytokinin receptor (green) and ER-marker p24δ5-RFP (red) in LRC cells (**a**, **b**) and root meristematic epidermal cells (**c–e**). Red arrowheads mark areas of co-localization. Fluorescence intensity profiles of the ER marker (red line) and CRE1/AHK4-GFP (green line) (**b**, **e**) were measured along the white lines (**a**, **c**) starting from upper end (0 μm) towards the arrowhead. Peaks of p24δ5-RFP fluorescence maxima at the endoplasmic reticulum overlap with CRE1/AHK4-GFP signal maxima in the LRC cells and root meristematic epidermal cells (black arrows; **b**, **e**). Peak of CRE1/AHK4-GFP non-overlapping with peaks of p24δ5-RFP in the root meristematic epidermal cells indicates localization at the plasma membrane (black arrowhead; **e**). Detailed view (**d**; white arrows point to CRE1/AHK4-GFP signal at the area of the PM). Scale bars = 5 μm.

real-time monitoring of the CRE1/AHK4-GFP in developing lateral root primordia (LRP). Although expression of CRE1/AHK4-GFP driven by *35S* promoter in LRP was relatively weak, similarly to cells in the root meristem, the CRE1/AHK4-GFP tends to localize at the ER and the PM (Supplementary Fig. 6h). Furthermore, in actively dividing cells we could detect a weak CRE1/AHK4-GFP signal during cell plate formation (Supplementary Movie 1).

Unlike cells located at the root apical meristem, in the differentiated cells of the LRC the CRE1/AHK4-GFP was detected in the ER, but not at the PM. To support further our conclusion about dominant localization of the cytokinin receptor at the ER in differentiated cells, we performed detailed observations of CRE1/AHK4-GFP in differentiated root epidermal cells above the meristematic zone. In these cells, the CRE1/AHK4-GFP was located at the ER (Supplementary Fig. 6i, j), but no co-localization with the PM reporter NPSN12 could be detected (Supplementary Fig. 6k, l).

Based on these observations we hypothesize that CRE1/AHK4-GFP located either at the ER or at the PM might activate distinct branches of downstream signalling to control specific process in differentiated versus meristematically active cells. Internalization and recycling of the receptor between PM and endosomal compartments in meristematic cells may represent another level in controlling signalling receptor function. Whether similarly to the CRE1/AHK4-GFP, also AHK2 and AHK3 might enter secretory pathway and reach the PM in meristematically active cells remains to be addressed. In previously reported studies localizations of the AHK3-GFP and AHK2-GFP have been observed in above-ground plant parts using transiently transformed *Nicotiana benthamiana* epidermal leaf cells[2] and transiently transformed Arabidopsis cotyledon cells[3], all in the differentiated stages. Hence, whether in specific cell types AHK2 and AHK3 might localize to the PM needs to be examined.

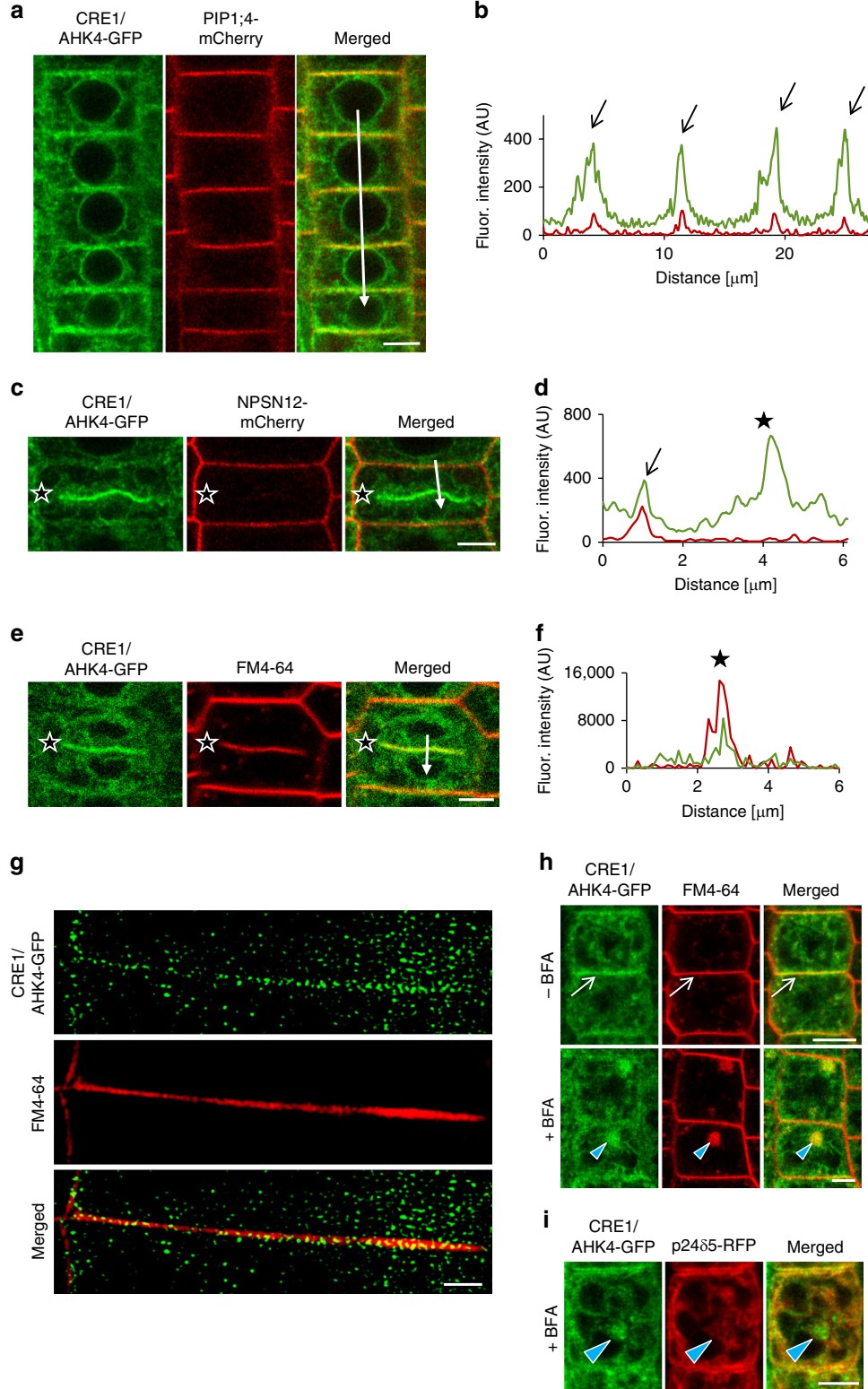

**Fig. 4 CRE1/AHK4-GFP co-localization with the plasma membrane markers in Arabidopsis root cells. a–f** Co-localization of CRE1/AHK4-GFP with the plasma membrane markers PIP1;4-mCherry (**a**, **b**), NPSN12-mCherry (**c**, **d**) and FM4-64 (**e**, **f**) in the root meristematic epidermal cells. Profiles of fluorescence intensity of PM marker (red line) and CRE1/AHK4-GFP (green line) (**b**, **d**, **f**) were measured along the white lines (**a**, **c**, **e**) starting from the upper end (0 μm) towards the arrowhead. Peaks of PIP1;4-mCherry and NPSN12-mCherry fluorescence maxima correlate with the plasma membrane staining and overlap with CRE1/AHK4-GFP signal maxima (black arrows) (**a–d**). CRE1/AHK4-GFP signal detected at the cell plate of dividing cell (black stars on **c**, **e**) co-localizes with FM4-64 marker (**e**, **f**; black stars) but not with the plasma membrane marker NPSN12-mCherry (**c**, **d**; black stars). **g** Super-resolution imaging (SIM) of CRE1/AHK4-GFP subcellular co-localization with FM4-64 labelled PM. **h**, **i** Co-localization of CRE1/AHK4-GFP and FM4-64 (**h**), but not p24δ5-RFP (**i**) in endosomal compartments (blue arrowheads) formed in root meristematic epidermal cells treated with 50 μM in BFA for 1 h. Note CRE1/AHK4-GFP localization at the plasma membrane (white arrows) prior BFA treatment. Scale bars = 5 μm (**a**, **c**, **e**, **h**, **i**) and 2 μm (**g**).

Taken together, monitoring of intracellular localization of the fluorescent cytokinin probe iP-NBD with higher affinity to CRE1/AHK4 cytokinin receptor, as well as direct visualization of CRE1/AHK4-GFP leads us to the conclusion that besides ER, cytokinin signal might also be perceived at other cellular compartments including the PM. As suggested by different localization of CRE1/AHK4 receptor in differentiated cells of LRC when compared to epidermal cells of root apical meristem, perception of cytokinin at either ER or PM might be cell- and developmental-context dependent. In particular, the strong expression of the cytokinin sensitive reporter *TCS::GFP* detected in columella and LRC cells[27] suggests that the ER-located cytokinin receptors activate cytokinin signalling cascade in these particular cell types. On the other hand, it remains to be resolved whether there is a specific branch of cytokinin signalling activated by receptors located at the PM of meristematic cells.

## Methods

**Plant material**. *Arabidopsis thaliana* (L.) Heynh (Arabidopsis) plants were used. The transgenic lines have been described elsewhere: *cre1-2*[28], *TCSn::ntdT:tNOS-pDR5v2::n3GFP*[16], *pup14*[5], p24δ5-RFP[29], HDEL-RFP[30], Wave lines 2R/RabF2b (LE/PVC), 9R/VAMP711 (Vacuole), 13R/VTI12 (TGN/EE), 18R/Got1p (Golgi), 22R/SYP32 (Golgi), 34R/RabA1e (Endosomal/Recycling endosomal), 131R/NPSN12 (PM) and 138R/PIP1;4 (PM)[20], *ARR5::GUS*[23], *35S::GFP* line was kindly provided by Shutang Tan (IST Austria, Austria). *35S::CRE1/AHK4-GFP* plants were generated as described below.

**Growth conditions**. Surface-sterilized seeds of Arabidopsis ecotype Columbia (Col-0) and the other transgenic lines were plated on half-strength (0.5×) Murashige and Skoog (MS) medium (Duchefa) with 1% (w/v) sucrose and 1% (w/v) agar (pH 5.7). The seeds were stratified for 2–3 days at 4 °C in darkness. Seedlings were grown on vertically oriented plates in growth chambers at 21 °C under long day conditions (16 h light and 8 h darkness) using white light (W), which was provided by blue and red LEDs (70–100 µmol m$^{-2}$ s$^{-1}$ of photosynthetically active radiation), if not stated otherwise.

**Pharmacological treatments for bio-imaging**. Seedlings 4–5-day-old were transferred onto solid 0.5× MS medium with or without the indicated chemicals. The drugs and hormones used were: $N^6$-benzyladenine (BA) in different concentrations (0.1, 0.5, 1 and 2 µM), *trans*-zeatin (tZ 0.1 and 1 µM), $N^6$-isopentenyladenine (iP, 5 µM). Mock treatments were performed with equal amounts of solvent (dimethylsulfoxide (DMSO)). Treatments with 5 µM iP-NBD and 5 µM Ade-NBD were performed in liquid 0.5× MS medium and imaging was carried out within 30 min time frame. For co-localization of the cytokinin fluoroprobe with PM marker, seedlings were pre-treated with 2 µM FM4-64 for 5 min and transferred into 5 µM iP-NBD supplemented 0.5× MS medium, followed by imaging within 30 min time frame. To explore affinity of iP-NBD to BFA endosomal compartments, seedlings were incubated in 50 µM BFA for 1 h and afterwards transferred into iP-NBD supplemented medium and imaged. Localization of CRE1/AHK4-GFP in BFA endosomal compartments was examined in 4–5-day-old seedlings incubated in 50 µM BFA for 1 h. For BFA washout experiments, seedlings were placed in a fresh BFA-free 0.5× MS medium, which was replaced every 10 min for at least 1 h.

**Recombinant DNA techniques**. The coding region of the cytokinin receptor CRE1/AHK4 (*At2g01830*) was amplified without the stop codon by PCR using a gDNA from *Arabidopsis thaliana* Col-0 as a template and cloned into the Gateway vector pENTR_2B dual selection (primers: AHK4_Fw_SalI_KOZAK *CGCGTC GACccaccATGAGAAGAGATTTTGTGTATAATAATAATGC* and AHK4_R_NotI *TTTTCCTTTTGCGGCCGCgaCGACGAAGGTGAGATAGGATTAGG*). To construct C-terminal fusion of CRE1/AHK4 with GFP, CRE1/AHK4 was shuttled into the destination vector pGWB5[31] containing *35S* promoter to create *35S::CRE1/AHK4-GFP* construct. For the transient Luciferase assay in Arabidopsis protoplasts, CRE1/AHK4-GFP fusion construct was re-cloned into p2GW7,0 vector. CRE1/AHK4-GFP region was amplified by PCR using pGWB5_CRE1/AHK4-GFP as a template (primers: *35S_FW CCACTATCCTTCGCAAGACCCTTC* and AHK4_5A_NheI_RE *TATTCCAATgctagcTTACTTGTACAGCTCGTCCATGC*) and ligated into the Gateway pENTR_2B dual selection entry vector. CRE1/AHK4-GFP was shuttled into the destination vector p2GW7,0.

**Plant transformation**. Transgenic Arabidopsis plants were generated by the floral dip method using *Agrobacterium tumefaciens* strain GV3101[32]. Transformed seedlings were selected on medium supplemented with 30 µg mL$^{-1}$ hygromycin.

**Homology modelling and molecular docking**. CRE/AHK structures were modelled based on CRE1/AHK4-iP crystal structure (PDBID: 3T4L)[12] using Modeller 9.10[33]. The geometry of iP-NBD was modelled with Marvin (http://www.chemaxon.com), and then the compounds were prepared for docking in the AutoDockTools program suite[34]. The Autodock Vina program[35] was used for docking iP-NBD ligand into the set of AHK structures obtained from homology modelling. A 15 Å box centred at the original ligand binding position was used. The exhaustiveness parameter was set to 20.

**Competitive binding assay in *E. coli* strains**. Receptor direct binding assays were conducted using the *E. coli* strain KMI001 harbouring either the plasmid pIN-III-AHK4 or pSTV28-AHK3, which express the Arabidopsis histidine kinases CRE1/AHK4 or AHK3[36,37]. Bacterial strains were kindly provided by Dr. T. Mizuno (Nagoya, Japan). The competitive binding assays[15,38] were performed with homogenous bacterial suspension with an OD$_{600}$ of 0.8 and 1.2 for CRE1/AHK4 and AHK3 expressing strains, respectively. The competition reaction was allowed to proceed with 3 nM [2-$^3$H]tZ and 6–18 nM [2-$^3$H]iP, and various concentrations of iP and iP-NBD, 0.1% (v/v) DMSO was added as a solvent control. After 30-min incubation at 4 °C, the sample was centrifuged (6000 × *g*, 6 min, 4 °C), the supernatant was carefully removed, and the bacterial pellet was resuspended in 1 ml of scintillation cocktail (Beckman, Ramsey, MN, USA) in an ultrasonic bath. Radioactivity was measured by scintillation counting by a Hidex 300 SL scintillation counter (Hidex, Finland). To discriminate between specific and nonspecific binding, a high excess of unlabelled natural ligand *tZ*, or iP (at least 3000-fold) was used for competition. The functional inhibition curves were used to estimate the IC$_{50}$ values. The $K_i$ values were calculated using the equation $K_i = IC_{50}/(1 + [radio-ligand]/K_D)$ according to Cheng and Prusoff[39]. [2-$^3$H]tZ and [2-$^3$H]iP were provided by Dr. Zahajská from the Isotope Laboratory, Institute of Experimental Botany, Czech Academy of Sciences.

**Quantitative RT–PCR**. RNA was extracted with Monarch® Total RNA Miniprep Kit (NEB) from roots of 5-day-old plants that were sprayed with mock (DMSO) or 0.01 µM iP, 0.1 µM iP, 1 µM iP, 10 µM iP-NBD, 100 µM iP-NBD or co-treatment of 0.1 µM iP + 10 µM iP-NBD, 0.1 µM iP + 100 µM iP-NBD for 15 min (Fig. 1d); Mock (DMSO) or 5 µM iP (Supplementary Fig. 5e); for 15 min. Poly(dT) cDNA was prepared from 1 µg of total RNA with the iScript cDNA Synthesis Kit (Bio-Rad) and analysed on a LightCycler 480 (Roche Diagnostics) with the Luna® Universal qPCR Master Mix (NEB) according to the manufacturer's instructions. The expression of CRE1/AHK4 of the two independent lines was quantified either with a specific primer pair (AHK4-GFP_FW: *TATCTCACCTTCGTCGTCGC* and AHK4-GFP_RE: *CCTTGCTCACCATGGATCCTC*) and their relative expressions were compared to the house-keeping gene PP2A (PP2A_FW: *TAACGTGGC-CAAAATGATGC* and PP2A_RE: *GTTCTCCACAACCGCTTGGT*) or with AHK4_FW: *GAACTGGGCACTCAACAATCA* and AHK4_RE: *ACGAATTCA-GAGCACCACCA* pair of primers and their relative expression refer to the Col-0 mock treatment. All qRT-PCR quantifications were done using PP2A as a reference gene (Fig. 1d; Supplementary Fig. 5a, e). For the *ARR* expressions (ARR5_FW: *TGCCTGGGATGACTGGATATG*, ARR5_RE: *CTCCTTCTTCAAGACATCTATCG*, ARR7_FW: *TACTCAATGCCAGGACTTTCAGG*, ARR7_RE: *TCTTTGAGA-CATTCTTGTATACGAGG*, ARR16_FW: *CGTAAACTCGTTGAGAGGTTGCTC* and ARR16_RE: *GCATTCTCTGCTGTTGTCACTTTG*), the fold change refer to the Col-0 mock treatment. The experiment was performed in three technical and three biological replicates.

**Measurements of iP-NBD cell transport kinetics**. Seedlings of 4-day-old Arabidopsis Col-0 were pre-treated for 20 min with 5 µM iP or DMSO and transferred into MS media containing 5 µM iP-NBD and 5 µM iP/DMSO and instantly imaged. To examine PUP14-dependent iP-NBD transport kinetics, 4-day-old seedlings of *pup14* and Col-0 were treated with 5 µM iP-NBD and immediately imaged. For both experiments imaging was performed in the same area of the root for 12 min every 2, 7 and 12 min to minimize photobleaching. iP-NBD fluorescence was measured with ImageJ in the LRC cells (iP pre-treatment experiment) and in the root epidermal cells (*pup14* experiment) from 4 to 7 cells originating from 4 to 5 roots.

**Imaging**. For confocal microscopy imaging, a vertical-stage laser scanning confocal Zeiss 700 (LSM 700) and Zeiss 800 (LSM 800), described in ref.[40], with a ×20/0.8 Plan-Apochromat M27 objective, an LSM 800 inverted confocal scanning microscope Zeiss, with a ×40 Plan-Apochromat water immersion objective and a Zeiss LSM 880 inverted fast Airyscan microscope with a Plan-Apochromat ×63 NA 1.4 oil immersion objective were used. Samples were imaged with excitation lasers 488 nm for GFP (emission spectrum 490–560 nm) and NBD (emission spectrum 529–570 nm), 555/561 nm (inverted/vertical) for RFP (emission spectrum 583–700 nm), FM4-64 (emission spectrum 650–730 nm), mCherry (emission spectrum 570–700 nm) and tdTomato (emission 560–700 nm).

For super-resolution SIM microscopy, an Axioimager Z.1 with Elyra PS.1 system coupled with a PCO.Edge 5.5 sCMOS camera was used. Samples were excited with the 488 nm and 561 nm laser lines. Oil immersion objective (×63/1.40) and standard settings (the grating pattern with five rotations and five standard

phase shifts per angular position) were used for image acquisition. Image reconstruction was done in Zeiss Zen software (black version with structured illumination module) using manual mode with adjusting the noise filter and super-resolution frequency weighting sliders as described in ref. [41]. For image post-processing, profile measurements and co-localization analysis, the Zeiss Zen 2011, ImageJ (National Institute of Health, http://rsb.info.nih.gov/ij), Photoshop 6.0/CS, GraphPad Prism 8 and Microsoft PowerPoint programs were used. For SIM co-localization experiments 30 PM regions originating from root cells of 5 seedling plants were used.

**Statistics**. The statistical significance was evaluated with the Student's *t* test and two-way ANOVA.

**Reporting summary**. Further information on research design is available in the Nature Research Reporting Summary linked to this article.

## Data availability

All data in this study are available in the main text or the Supplementary materials. Extra data are available from the corresponding authors upon request. Source data are provided with this paper.

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

## Acknowledgements

This paper is dedicated to deceased P. Galuszka for his support and contribution to the project. This research was supported by the Scientific Service Units (SSU) of IST-Austria through resources provided by the Bioimaging Facility (BIF), the Life Science Facility (LSF) and by Centre of the Region Haná (CRH), Palacký University. We thank Lucia Hlusková, Zuzana Pěkná and Martin Hönig for technical assistance, and Fernando Anieto, Rashed Abualia and Andrej Hurný for sharing material. The work was supported from ERDF project "Plants as a tool for sustainable global development" (No. CZ.02.1.01/0.0/0.0/16_019/0000827), from Czech Science Foundation via projects 16-04184S (O.P., K.K. and K.D.), 18-23972Y (D.Z., K.K.), 17-21122S (K.B.), Erasmus+ (K.K.), Endowment Fund of Palacký University (K.K.) and EMBO Long-Term Fellowship, ALTF number 710-2016 (J.C.M.); People Programme (Marie Curie Actions) of the European Union's Seventh Framework Programme (FP7/2007-2013) under REA grant agreement no. [291734] (N.C.); DOC Fellowship of the Austrian Academy of Sciences at the Institute of Science and Technology, Austria (H.S.).

## Author contributions

L.S., O.P., E.B., P.G. and M.S. conceived the project; K.K. and J.C.M. performed most of the microscopic and biochemical experiments; O.Š. and J.Š. participated in plasma membrane localization studies and helped with evaluation and interpretation of sub-cellular localization data; H.S. contributed to generation and characterization of CRE1/AHK4-GFP transgenic lines; N.C. established the *TCS::LUCIFERASE* assays to test CRE1/AHK4-GFP activity in protoplasts; P.M. contributed to the *TCS* experiment in

planta; L.P., K.D. and V.M. designed and chemically synthesized cytokinin fluorescent probe; J.N. performed cytokinin in vitro assays; O.N. conducted the purification and quantification of cytokinins; K.B. performed in silico docking experiments; D.Z. analysed and interpreted the data; L.S., E.B. and O.P. designed experiments, analysed and interpreted the data; K.K., J.C.M., O.Š. and L.S. made the figures; E.B., O.P. and L.S. wrote the paper.

## Competing interests

The authors declare no competing interests.
