## [Peer Review File · Nature Communications]

Reviewers' comments:

Reviewer #1 (Remarks to the Author):

There has been debate about how CK signaling or the TCS pathway functions, since the discovery of the CK AHK receptor localization primarily to the ER membrane, rather than the PM where it was previously thought to solely exist. Interestingly, every report of AHK localization to the ER, still indicates that some AHKs are present at the PM. These reports re-verifies that AHKs would appear to be at the ER membrane, but also in some cells also the plasma membrane. The question then becomes, how important is this and is the occurrence of an AHK receptor on the PM an oddity of some root cells or a more general norm, which occurs in many cell types throughout the plant, likely beyond the scope of this work. This study confirms that a CK receptor can be at the PM and minimally trigger a proxy for CK response in the TCS-GFP reporter.

More specifically, the focus of this manuscript is to show that the (CK) cytokinin AHK receptors are not solely at the ER membrane, but can also exist in some cell types at the plasma membrane (PM). Assume that all is correct from the work presented, it would seem that the author have achieved this. This is accomplished using an AHK4-GFP line and a fluorescently labeled CK (iP-NBD) and confocal examination of each of these in roots cells along with a number of other fluorescent organelle markers to confirm position. The different localization patterns of AHK4 largely in the ER of LRC, while seemingly in both ER and PM of epidermal cells, is quite interesting. This would potentially suggest possible cell type specification of CK receptors in regards to localization to the PM, while ER localization stays more universal.

Points that should be addressed in this manuscript going forward.

Most of the work in this manuscript is based on a fluorescence CK and a transgenic cytokinin receptor AHK4-GFP line and there are things with each of these that should be expanded upon in this manuscript

1. Since most of the findings in this manuscript are with the labeled iP-NBD, why is iP not the active CK control in everyone of the assays performed. This makes it difficult to directly compare how well iP-NBD is really functioning. Especially since there are known differences in most of these same assays between different CK forms (iP, tZ, BA)
 - a. Competitive binding assay Fig 1c
 - b. ARR5-GUS activity assay Fig 1d
 - c. Receptor activity assay Fig S1a, f
 - d. TCS-GFP activity Fig S1b
2. Please discuss how iP-NBD fits into the receptor binding pocket when other N-9 conjugated compounds have been reported as not fitting. Since the binding pocket fit is just hypothetical, shouldn't AHK2 should be modeled also.
3. Some data on iP-NBD stability or form change needs to be added after applying this chemical to plants, particularly Arabidopsis roots or the major cell types examined here. All that is shown is reduced CKX2 activity, when other CKXs might fully affect this compound, possibly in an organelle specific manner, as CKXs are known to do. There is also a % normalized content shown that indicates a rapid change in content that should be addressed with this. Also, while this compound cannot be O-conjugated or N9 conjugated, since the linker is on the N9 position precluding that, why as stated in Ln 65-67 could this not be conjugated at the N7 position? The labs involved have the ability to check any changes that might occur with this compound in plants during the lifetime of the experiments presented.
4. The 35S::AHK4-GFP line generated in this manuscript must be described in much greater detail, including level of AHK4 expressed in this overexpression line. Since there is a root growth defect this is problematic in the consideration of all information generated from it. Are those two independently selected lines in Fig S4 showing all the same features. This should really be in an ahk4 mutant background to show that this construct is able to functionally complement this mutation as well as keep receptor levels normal-ish in this line. The only other example I can find in the literature of an AHK line being overexpressed is for AHK3 done with a GFP tag on both the N

and C terminus, but that line was only generated in an *ahk2,3* DM background, which in complemented (Ceaser et al 2011: JXB). As such it is hard to know if this reporter line is functioning in a normal manner. This must be addressed.

5. The finding of change in TCS:LUC sensitivity in protoplasts (Fig S4a) just indicates that there is an altered sensitivity. If that is because there is too much AHK4 made in that background, then that is a big issue. If there is too much AHK4 (from abnormal overexpression) on the ER, could it be placed on the PM in some cells? Basically, is an unnatural state reached and the output of what is seen, just misplacement of too much AHK4?

6. The level of iP-NBD used in the TCS-GFP assay FigS1 is very low compared to what is used elsewhere in this manuscript. Also the 15h treatment in that assay, again raises issues about breakdown or conjugation.

7. In Figure S3, please change the white arrows to something different in either (a) or (b-e) to help differentiate these, which are indicating quite different things.

8. There should be some mention of pH and receptor binding somewhere in the text, as this has been raised as an issue before in relation to the acidity of different organelles.

9. What happens in the other basic cell types in the root? There was a lot of imaging in epidermal and LRC cells, but what about other cell types that are very nearby in the root? It would be nice to indicate if there is a general status of mostly being ER localized in the root, as had been seen before in leaf cells, and the epidermis is the exception, or does every cell type need a serious investigation to determine this and it may be very cell type dependent?

Reviewer #2 (Remarks to the Author):

The authors use two independent approaches to address a long standing question of the field: what are the cellular compartments from which cytokinin signaling is initiated? Signaling from the ER has been postulated based on the observations that the majority of receptors seems to localize to the ER membranes, with their sensing domain facing the lumen. However, receptors have also been observed at the PM, and based on recent work with the cytokinin transporter PUP14 the apoplast was proposed to be important for signal initiation.

First, the authors use a fluorescently labeled cytokinin (iP-NBD) as a tool to study the subcellular localization of cytokinin-receptor complexes in different cell types.

The attachment of the fluorochrome to the cytokinin iP reduces the affinity between ligand and AHK3, or AHK4 receptors about 100 fold. Also, the modified ligand is unable to initiate a signaling response. Despite these limitations, the authors could show by competition assays, that (iP-NBD) binding is specific and therefore, the fluorescence emitted by is indicative of cytokinin subcellular localization.

In addition, they study the subcellular localisation of the AHK4 cytokinin receptor, translationally fused to GFP. Importantly, they could show the functionality of the used transgene as it was functional in planta in absence the endogenous AHK4.

Extensive studies with different treatments and co-stainments with different cellular markers were performed and revealed that receptors and receptor-ligand complexes are found in the endoplasmatic reticulum. In epidermal cells of the root meristem, receptors and receptor-ligand complexes were additionally shown to enter the endomembrane trafficking system. The AHK4-GFP protein was also detected at the plasma membrane and at newly forming cell plates, supporting the possibility that AHK4 at the PM may be involved in signaling initiation. Together with the data presented in the parallel submitted manuscript, the existence of cytokinin-signaling initiation from the PM, in particular mediated by AHK4, is convincingly established.

This work presented here is of high quality work at all levels. Keep in mind that the questions addressed are technically very difficult, as cytokinin ligands and cognate receptors are basically impossible to detect in situ. Therefore, the authors need to use somewhat imperfect tools: iP-NBD has lower affinity to receptors and does not initiate signaling, 35S::AHK4-GFP does show some overexpression phenotypes. However, with all controls and given the fact that both approaches

independently provide similar results, the overall conclusions are well supported.

I have only one question, or suggestion: I would expect iP-NBD staining in an *ahk4* and/or *ahk3/4* mutant could be a good negative control, and it would be helpful to perform and present that data.

Reviewer #3 (Remarks to the Author):

In this manuscript, the authors developed a fluorescently labeled cytokinin probe (iP-NBD), and analyzed subcellular localization of the probe and GFP-fused AHKs, specially focusing on CRE1/AHK4. They found action of the compound on the receptor as an antagonist having higher affinity to CRE1/AHK4 than other receptors. Then, the authors carefully analyzed subcellular localization of the fluorescent probe and CRE1/AHK4-GFP in lateral root cap cell and epidermal cell at root meristematic zone. Based on the observed results, they suggested that CRE1/AHK4 can enter secretory pathway via Golgi body and reach the plasma membrane. They also observed localization of iP-NBD, and suggested cytokinin could be perceived both ER and plasma membrane. This study provides some important information for cytokinin perception at plasma membrane and ER. Especially, several important findings were shown for deeper understanding of intracellular traffic of cytokinin receptor. On the other hand, most of the data for "functionality" of the receptors are indirect. I raised several points to be addressed.

1: Title is not attractive, but just descriptive.

2: The authors used benzyladenine, an artificial cytokinin, to show antagonistic effect of iP-NBD (Fig 1d). I strongly suggest the authors to use authentic cytokinins, such as iP. In other experiments, they used authentic ones. I would like to know why they used BA in this experiment.

3: Line 73, the authors described K_i values of iP-NBD for AHK3 ($\sim 37 \mu\text{M}$) and CRE1/AHK4 ($\sim 1.4 \mu\text{M}$). But it is difficult to see the values from Fig 1c, especially AHK3 + iP-NBD.

4: Information of iP and iP-NBD concentrations is missing in Fig 1e.

5: The subcellular localization pattern of AHK-GFP fluorescence was relatively clear, whereas that of iP-NBD fluorescence was not. The fluorescence appears to be uniformly distributed. To show the specific binding, they showed Ade-NBD in lateral root cap cells as a control in Fig S1g, but more information, such as the data of epidermal cells should be provided. Therefore, it is not sure whether the overlap with the subcellular localization pattern of other marker proteins is specific event or not.

Reviewer #4 (Remarks to the Author):

Cytokinins are essential plant growth phytohormones which promote plant cell growth, division and differentiation as well as various developmental and physiological processes. In this study, Kubiasová et al., have developed a useful cytokinin fluorescent probe termed iP-NBD with higher affinity to CRE1/AHK4 cytokinin receptor. By using the probe, the authors further demonstrated that the cytokinin receptor CER1/AHK4 participates in the endomembrane trafficking system from ER to the plasma membrane, possibly involved in mediating cytokinin signaling and gradient in plant roots. In contrast to the conventional ER-localization of cytokinin receptor, demonstration of the new subcellular localization of CER1/AHK4 on the PM provides a new perspective of cytokinin signaling pathway. The findings of the study are novel and interesting to the field. In addition, it may help to understand of establishment of cytokinin response gradient in different plant organs since the detailed underlying mechanisms are not well understood. However, I would suggest the authors to address the following concerns:

1. The image quality in some of the Figures and Supplementary Figures is poor. The real fluorescent signal is difficult to be recognized in the high noise background. Meanwhile, such images are neither representative nor convincing enough to support the conclusions as stated in the manuscript. For example, the iP-NBD background noise in Fig. 2b is too high to separate the real signal and noise. It is obviously not convincing enough to demonstrate that iP-NBD localizes in ER although ER specific marker was used for the colocalization study. The ER marker showed the typical ER patterns but the iP-NBD did not. Similar problems also can be found in Fig.S3b, c and d.
2. For all of the colocalization and non-colocalization studies in all of the Figures, the authors are supposed to perform quantitative and statistic analysis to clearly demonstrate the ratio of the colocalization. It is especially useful for illustration of the cases of partial colocalization and non-colocalization of two different organelles/vesicles, such as in Fig. 2, Fig. S2a, Fig.3 and Fig. S3.
3. The time of FM4-64 endocytosis should be clarified in the text and figure legend since the FM4-64 is firstly endocytosed to the early endosome (TGN) and then goes to the late endosome (MVB), and if given enough time, it will eventually reach the tonoplast of the vacuole. What are the FM4-64 cytosolic dots in Fig 2 mainly representing for?
4. The conclusion stated in the text line 133-136 can not be fully supported by Fig.2 c,e and Fig. S2b. The central vacuole in plant cells is usually large and it pushes the other organelles and cytosol to the cortical areas which are underneath the PM. Therefore, how to rule out the possibility that the PM pattern highlighted by iP-NBD is actually from ER because iP-NBD in the ER is pushed very close to the PM by the central vacuole, as shown in Fig. 2c and e? Additionally, the florescent intensity analysis as shown in Fig. 2d and f can not support the PM-localization of iP-NBD in plant cells.
5. The authors chose two types of plant cells: LRC and epidermal cell as the studying materials. What are the reasons for selecting them?
6. The subcellular localization pattern of cytokinin receptor CRE1/AHK4 in two different plant cell types are different (one is mainly on ER and the other one on both ER and PM), however, there is no further discussion to illustrate the possible reasons. Except for CRE1/AHK4, how about the other different cytokinin receptors (also localized to PM or depending on different cell types)?
7. Cytokinin mainly consists of two active types: iP and tZ. In this study, the author use iP-NBD as a cytokinin reporter. What about tZ? Do the two forms of cytokinin show the similar or different subcellular localization patterns? Compared to the results obtained from Antoniadin et al., how to explain and make sense the differences in functional roles and responses of iP and tZ in the two independent studies?

Response to Reviewers' Comments

We thank the reviewers for their valuable comments and suggestions. We have expanded the existing results and incorporated new observations into the revised version of our manuscript, which we hope address and resolve all major concerns of reviewers.

We have updated our revised manuscript with a substantial number of new results and extended discussion. Previous Fig. 3 was updated and now appears as a new main Fig. 4, Fig. 3 newly shows overview of CRE1/AHK4-GFP signal distribution in the root meristematic zone. All Supplemental figures were updated with better quality confocal images and new experimental data.

Briefly, for the new results we have:

- 1. Docking simulation of AHK2 receptor ligand binding alongside AHK3 simulation using the CRE1/AHK4-iP crystal structure, and a new competitive binding assay using [2-³H]iP as a competitor, in updated Fig. 1.*
- 2. Updated Fig. 2 (panels a,b) and a new Fig. 3 and Fig. S6, which now include analyses of CRE1/AHK4-GFP colocalization with the plasma membrane and ER reporters in the different cell types (including lateral root cap cells, epidermal cells in meristematic and differentiation zone, provascular cells and lateral root primordium).*
- 3. Quantification of iP-NBD response using new pTCSn::ntdTomato cytokinin reporter in Supplemental Fig. S1, expression analyses of CRE1/AHK4-GFP by qRT-PCR and Western blot and extended phenotype characterization of CRE1/AHK4-GFPox lines including expression analyses of early cytokinin response genes and root growth inhibition by exogenous cytokinin in individual 35S::CRE1/AHK4-GFP lines in Supplemental Fig. S5.*
- 4. Co-localization analysis of iP-NBD with marker lines for diverse compartments presented in Supplemental Fig. S3 and evaluated as the Pearson coefficients in Table 1.*

Below we directly respond to issues raised by reviewers point by point. Changes to the main text incorporated into the revised manuscript are in red font in this document.

Reviewers' comments:

Reviewer #1 (Remarks to the Author):

There has been debate about how CK signaling or the TCS pathway functions, since the discovery of the CK AHK receptor localization primarily to the ER membrane, rather than the PM where it was previously thought to solely exist. Interestingly, every report of AHK localization to the ER, still indicates that some AHKs are present at the PM. These reports re-verifies that AHKs would appear to be at the ER membrane, but also in some cells also the plasma membrane. The question then becomes, how important is this and is the occurrence of an AHK receptor on the PM an oddity of some root cells or a more general norm, which occurs in many cell types throughout the plant, likely beyond the scope of this work. This study confirms that a CK receptor can be at the PM and minimally trigger a proxy for CK response in the TCS-GFP reporter.

More specifically, the focus of this manuscript is to show that the (CK) cytokinin AHK receptors are not solely at the ER membrane, but can also exist in some cell types at the plasma membrane (PM). Assume that all is correct from the work presented, it would seem that the author have achieved this. This is accomplished using an AHK4-GFP line and a fluorescently labeled CK (iP-NBD) and confocal examination of each of these in roots cells along with a number of other fluorescent organelle markers to confirm position. The different localization patterns of AHK4 largely in the ER of LRC, while seemingly in both ER and PM of epidermal cells, is quite interesting. This would potentially suggest possible cell type specification of CK receptors in regards to localization to the PM, while ER localization stays more universal.

Points that should be addressed in this manuscript going forward.

Most of the work in this manuscript is based on a fluorescence CK and a transgenic cytokinin receptor AHK4-GFP line and there are things with each of these that should be expanded upon in this manuscript

1. Since most of the findings in this manuscript are with the labeled iP-NBD, why is iP not the active CK control in everyone of the assays performed. This makes it difficult to directly compare how well iP-NBD is really functioning. Especially since there are known differences in most of these same assays between different CK forms (iP, tZ, BA)

- a. Competitive binding assay Fig 1c
- b. ARR5-GUS activity assay Fig 1d
- c. Receptor activity assay Fig S1a, f
- d. TCS-GFP activity Fig S1b

***Response:** We thank the reviewer for insightful and helpful comments. In some assays we used BA as an active form of cytokinin, which exhibits a good stability in biological assays, however we agree with the criticism that in the cases when comparisons of iP-NBD vs CKs are made, BA might not be the most suitable positive control. As suggested, we performed the requested analyses using iP as an active form of cytokinin throughout all presented bioassays.*

Ad a) Competitive binding assays with AHK3 and CRE1/AHK4 were performed using tritium-labelled iP ([2-3H]iP; Fig. 1c and Fig. S1a):

Competitive binding assays with E. coli expressing either AHK3 or CRE1/AHK4 showed that iP-NBD competes for receptor binding with radiolabelled natural cytokinins iP and tZ in different ranges of ligand concentrations (Fig. 1c and Fig. S1a), corresponding with the receptor ligand preferences. As predicted, iP-NBD had lower affinity to AHK3 (with $K_i \sim 37 \mu\text{M}$ and $> 100 \mu\text{M}$ against radiolabelled tZ and iP, respectively) than to CRE1/AHK4 (with $K_i \sim 1.4 \mu\text{M}$ and $\sim 31 \mu\text{M}$ against radiolabelled tZ and iP, respectively), indicating that this fluoroprobe is more specific to CRE1/AHK4 (Fig. 1c and Fig. S1a).

The assays confirmed the previous results obtained with [2-3H]tZ. iP-NBD blocked the binding of [2-3H]iP in the concentration dependent manner in the case of CRE1/AHK4 reaching K_i of $31 \mu\text{M}$. The relatively higher value of K_i obtained with [2-3H]iP in the CRE1/AHK4 binding assay, compared to [2-3H]tZ results, probably reflects comparatively lower affinity of iP towards CRE1/AHK4 receptor (Romanov et al., 2006) – the concentration of [2-3H]iP competitor thus had to be increased compared to [2-3H]tZ assays to get statistically significant readouts. Similarly, we experienced the same problem with AHK3 receptor, where the difference between affinities of tZ and iP towards the receptor is even more pronounced (about 100-fold higher affinity of tZ over iP; Romanov et al., 2006). Therefore, no visible decrease of the bound radioactivity was recorded including in the highest concentration of iP-NBD, when [2-3H]iP was used as a natural ligand in the AHK3 binding assay. The

relatively low affinity of iP towards AHK3 means that iP-NBD, a derivative of iP, has even lower affinity to AHK3 receptor and much higher concentrations would be needed to record potential competition for binding. Further, from the technical point of view higher concentration of the radiolabeled iP is needed to achieve recording of reasonable affinity to AHK3 that is then, in turn, more difficult to compete with even less recognized ligand iP-NBD. Use of [2-3H]iP is thus logical when iP-NBD as an iP-derived compound is compared for its binding to a cytokinin receptor with its “parental” compound (a natural cytokinin), but fails to show the situation of a right positive control (that should be a preferred ligand) in a biochemical ligand binding experimental setup. For this reason, we kept the previous graph showing the competition of iP-NBD with radiolabeled tZ in Fig. S1 (Fig. S1a). In this case, using [2-3H]tZ should be interpreted to represent a better “positive” control for comparison of iP-NBD ability to interact with AHK3 and CRE1/AHK4 that both recognize tZ as a preferred ligand with the highest affinity.

Ad b) ARR5-GUS activity assay (new Fig. 1d) was repeated with Col-0/ARR5-GUS plants using iP as a reference cytokinin. This assay showed similar result to the previous one performed with BA, pointing to a weak activation of ARR5:GUS by the highest concentration of iP-NBD. 10 μ M iP-NBD caused induction of the cytokinin reporter comparable to 0.01 μ M iP. Due to the fact that the competitive assay has been now more thoroughly done (using 2 time-points) with more sensitive pTCSn::ntdTomato:TNOS cytokinin reporter (Fig S1c), combination of iP and iP-NBD was not further tested in Col-0/ARR5-GUS plants. However, we kept the graph in the new Fig. 1d to demonstrate the interaction of higher iP-NBD concentrations with cytokinin signaling using this traditional reporter. The effect of simultaneous application of iP and iP-NBD using the new TCS reporter (Fig. S1c) is further described below. The impact of iP-NBD on induction of ARR5-GUS in plants with different receptor mutant backgrounds was unfortunately not re-evaluated with iP as a control due to technical reasons. While it should be interesting to compare the effect of iP vs tZ exogenous treatments, current unavailability of CK receptor double mutant/ARR5::GUS lines in our laboratories as well as in the plant hormone community preclude us from performing these experiments in near future. However, in our opinion this experiment is not crucial for explanation of the interaction of iP-NBD and we believe that other strong evidence showing that CRE1/AHK4 directly interacts with iP-NBD fluoroprobe is provided.

Ad c) Receptor activity assays describing the ability of iP-NBD to activate receptors AHK3 and CRE1/AHK4 (previous Fig. S1a), and competitive binding assays showing affinity of Ade-NBD to AHK3 and CRE1/AHK4 (previous Fig. S1f) were re-evaluated based on comparison with iP (new Fig. S1b) and [2-3H]iP ligand competitions, respectively (new Fig. S2e).

Ad d) Instead of the previously used cytokinin reporter TCSn::GFP (previous Fig. S1b) we now present use of pTCSn::ntdTomato:TNOS, an improved version of a cytokinin sensitive reporter (De Smet et al., 2019) for iP-NBD activity inspection. This reporter with excitation at 561 nm and emission spectra at 560-700 nm, respectively, is more suitable for treatment with iP-NBD detected at 490-560 nm – see the new Supplemental Fig. S1c. Similarly to previous results using BA derivative, we confirmed that treatment with iP for 6 or 15 hours significantly increased expression of the cytokinin sensitive reporter. Application of iP-NBD did not significantly affect expression of the reporter, however when applied simultaneously, iP-NBD attenuated iP mediated enhancement of the TCS-driven reporter expression (Fig. S1c). This result is in good agreement with our previous results obtained with ARR5-GUS reporter showing effect of iP-NBD on the attenuation of the signal induced by 1 μ M BA. This is also expected taking into account the weak agonistic effect of iP-NBD recorded in Col-0/ARR5-GUS plants (Fig. 1d). Altogether, both the new and previously presented results suggest that iP-NBD directly interacts with the ligand binding domain (CHASE domain) of the cytokinin receptors

(primarily with the CRE1/AHK4 receptor) and plays a role of a partial agonist, i.e. compound that can bind to and activate a given receptor with only partial efficacy compared to the full agonist. Under the conditions of high ligand saturation, iP-NBD may also be acting as a competitive antagonist, competing with the full agonist for a receptor occupancy, which in turn leads to the attenuation of the cytokinin signaling output observed in pTCSn::ntdTomato:TNOS competitive assay.

2. Please discuss how iP-NBD fits into the receptor binding pocket when other N-9 conjugated compounds have been reported as not fitting. Since the binding pocket fit is just hypothetical, shouldn't AHK2 should be modeled also.

Response: Thank you for pointing this out. We have newly added also description of AHK2 heterologous docking simulation (Fig. 1b). The docking simulations showed that iP-NBD can be bound into the active sites of all the receptors with the binding energies ranging from -9,7 kcal/mol for AHK2 to -8,7 kcal/mol for AHK3 and CRE1/AHK4. The topology of iP and iP-NBD/receptor CRE1/AHK4 complex is similar in each of the cases:

Docking into the CRE1/AHK4-iP crystal structure showed that iP-NBD binds into the receptor cavity in a similar manner to iP, but the lack of interaction via N9 (which links the fluorescent probe) causes the purine ring shift leading to the larger distance and thus weaker interaction between N7 and Asp137 (Fig. 1b).

Recently, it was shown that cytokinin ribosides can be true ligands of the cytokinin receptors (Jaworek et al., 2019) and they also show true ligand binding properties with some CK sensitive Arabidopsis HKs (Romanov et al., 2006) as well as maize HK receptors (Yonekura-Sakakibara et al., 2004). The most typical N9-conjugated cytokinins, ribosides, have the ring structure directly conjugated to N9 of the adenine ring. As shown in the visualization of the CRE1/AHK4 binding site using known co-crystal structure with iP base (published by Hothorn et al., 2011) (Fig. 1b), entrance into the cavity is too narrow to enable cytokinin ribosides to interact with the receptor because the ribose ring would not fit the channel entrance without causing a dynamic change to allow such structure to be formed. Due to the fact that the co-crystal of a cytokinin receptor with a cytokinin riboside haven't been described yet, such a dynamic change needed can be only predicted theoretically. However, as already mentioned cytokinin ribosides have been shown to be true ligands in the functional assays and such dynamic change should be possible. In contrary to the cytokinin ribosides, iP-NBD uses a short ethylene linker that allows the fluorophore to fit into an antechamber of the receptor, whereas its cytokinin part occupies the same space as iP free cytokinin base (Fig. 1b).

3. Some data on iP-NBD stability or form change needs to be added after applying this chemical to plants, particularly Arabidopsis roots or the major cell types examined here. All that is shown is reduced CKX2 activity, when other CKXs might fully affect this compound, possibly in an organelle specific manner, as CKXs are known to do. There is also a % normalized content shown that indicates a rapid change in content that should be addressed with this. Also, while this compound cannot be O-conjugated or N9 conjugated, since the linker is on the N9 position precluding that, why as stated in Ln 65-67 could this not be conjugated at the N7 position? The labs involved have the ability to check any changes that might occur with this compound in plants during the lifetime of the experiments presented.

Response: Thank you for these advices and suggestions. Performing detailed metabolic analysis in different cell types may be possible (though technically rather challenging) but we do not find it necessary in the light of current facts and results we have already presented. Some of them we summarize in the following lines giving more explanations that might be missing in the previous manuscript due to the space limitations.

First, we refer to the chemical structure of iP-NBD. It was designed to minimize the possibility of metabolic conversion by cytokinin deactivation enzymes in planta:

The selection of the cytokinin fluorophore was done with intention to minimize the possibility of metabolic conversion by cytokinin deactivation enzymes in planta. For this reason, isopentenyladenine (iP) was used as a natural cytokinin that cannot be transformed through O-glycosylation at the cytokinin side chain. Furthermore, covalent attachment of 7-nitro-2,1,3-benzoxadiazole (NBD), a small fluorophore, to the N9 position of iP eliminates a risk of a metabolic conversion of the final cytokinin fluorescent probe iP-NBD (Fig. 1a) through N-glycosylation, or formation of cytokinin nucleotides. The stable attachment of the N9-substituent also prevents modifications at the N7 position by making this CK derivative completely inaccessible for N-glucosyltransferases.

It is well known that CKs can be effectively converted to their N7- and N9-glucosides by the action of N-glucosyltransferases (UGTs; Hou et al., 2004). In the case of iP-NBD the action of UGTs is completely eliminated through covalent attachment of 7-nitro-2,1,3-benzoxadiazole (NBD) to the N9 position of iP. The stable attachment of the N9-substituent prevents the glucosylation at this position. The chemical rearrangements (shift of the double bond between C8 and N7) caused by the substitution at N9 are making the N7 non-reactive as far as there is no hydrogen anymore to be substituted. Thus, NBD attachment further prevents any modifications in N7 position. Indeed, all the N9-substituted CKs and related structures could not be glucosylated by UGTs in the study of Hou et al. (2004). For this reason, chasing of any potential N7-metabolized iP-NBD structures does not probably bring any benefits (these compounds are only theoretically possible) and it also pose serious technical difficulties for their detection due to the current unavailability of the necessary control compounds.

Second, taking into account the chemical structure of iP-NBD that prevents O- and/or N-glycosylation the presumed in planta catabolic pathway of this molecule might be N6 side-chain cleavage by endogenous CYTOKININ OXIDASE/DEHYDROGENASE (CKX) activity. Using Arabidopsis cell line and detailed LC-MS/MS analysis we show (Fig. S2b) that deprenylation of iP-NBD is the main catabolic pathway leading to the formation of a CK inactive form, Ade-NBD, in a short time-scale (about 50% decrease in intracellular iP-NBD levels after 2 h treatment in planta):

Hence, the in vivo stability of the fluorophore was tested. iP-NBD was applied to Arabidopsis cells, and its intracellular processing was followed over a period of 0.5 - 5 h by LC-MS/MS analysis. Thus, iP-NBD and N9-NBD-labelled adenine (Ade-NBD), the expected product of iP-NBD deprenylation by CKXs, were used as molecular standards. Under these conditions, iP-NBD showed high stability within the first 30 minutes ($\geq 90\%$ recovery of intact molecule), dropping drastically after 5 hours. The concentration of Ade-NBD steadily increased, reaching the maximal concentration after 4 hours (Fig. S2b).

For this reason, we used short 5-15 min iP-NBD treatments/staining throughout the manuscript, when Ade-NBD intracellular levels should not be significant.

The CKX-mediated catabolic pathway observed *in vivo* suggested that iP-NBD can be recognized by CKXs as a substrate. To confirm this, we repeated and further described in the revised manuscript the enzymatic assay with recombinant AtCKX2, one of the most active CKX isoforms with apoplastic localization (Werner et al., 2003). AtCKX2 converts iP-NBD to the product with app. 6-times lower turnover rate (k_{cat}) compared to the parental iP molecule, but only with 33% lower catalytic efficiency (V_{max}/K_m). The detailed kinetic parameters are compared in Fig. S2c. We consider it likely that other AtCKX isoforms recognize iP-NBD as a substrate with different affinities, however, taking into account the exogenous application of iP-NBD, the apoplastic AtCKX2 is the most relevant CKX isoform to be tested. It is worth mentioning here that overexpressed AtCKX2 was shown to be efficient tool to deplete CK apoplastic levels leading to the attenuation of CK signaling (Zürcher et al., 2016).

Further, we newly added stability measurements of iP-NBD *in vitro* under different pH conditions that may occur in different organelles and in the apoplast (new Supplemental Fig. S2a). We found that even prolonged incubation of iP-NBD under acidic pH (pH 4 for 6 h) does not affect iP-NBD stability *in vitro*, and therefore, we can assume that iP-NBD is only metabolized in planta by means of enzymatic deactivation, preferentially by CKX action.

4. The 35S::AHK4-GFP line generated in this manuscript must be described in much greater detail, including level of AHK4 expressed in this overexpression line. Since there is a root growth defect this is problematic in the consideration of all information generated from it. Are those two independently selected lines in Fig. S4 showing all the same features. This should really be in an *ahk4* mutant background to show that this construct is able to functionally complement this mutation as well as keep receptor levels normal-ish in this line. The only other example I can find in the literature of an AHK line being overexpressed is for AHK3 done with a GFP tag on both the N and C terminus, but that line was only generated in an *ahk2,3* DM background, which in complemented (Ceaser et al 2011: JXB). As such it is hard to know if this reporter line is functioning in a normal manner. This must be addressed.

Response: We are very grateful for this suggestion and have performed expression analysis of CRE1/AHK4-GFP in two independent 35S overexpression lines using qRT-PCR and Western blot approaches to explore this. In the revised manuscript we show, that expression of CRE1/AHK4-GFP is increased when compared to the wild type control in both lines analyzed in this study (new Supplemental Fig. S5a-c).

We agree that for reliable expression and localization studies, a translational fusion of gene of interest with reporter driven by the endogenous promoter in a loss of function mutant is the most optimal. However, despite our efforts (several attempts of testing native promoter constructs), CRE1/AHK4-reporter fusion construct driven by an endogenous promoter was not expressed to the levels sufficient to perform the live cell imaging experiments. Therefore, we pursued an alternative strategy and generated lines expressing CRE1/AHK4-GFP driven by the constitutively active 35S promoter. As in all such cases, expression of the CRE1/AHK4-GFP translational fusion does not match its endogenous expression pattern and we therefore favored transformation of the wild type and testing for phenotype features related to enhanced cytokinin receptor activity. We analyzed the functionality of the CRE1/AHK4-GFP fusion in the protoplasts assay and showed that it significantly enhanced the expression of the TCS::LUC cytokinin sensitive reporter in response to cytokinin treatment (Fig. S5d).

In addition, we also analyzed expression of cytokinin early response genes including ARR5, ARR7 and ARR16 (type-A cytokinin response regulators) in AHK4-GFPox 5-day-old seedlings in response to cytokinin treatments.

Application of cytokinin resulted in strong upregulation of ARR5 and ARR7 cytokinin response factors in wild-type and both transgenic lines expressing CRE1/AHK4-GFP (Fig. S5e). However, a significantly enhanced transcription of ARR5 and ARR7 in response to cytokinin when compared to wild type was detected only in the line 35S::CRE1/AHK4-GFP (2), which displays a higher accumulation of CRE1/AHK4-GFP. ARR5 and ARR7 have been reported as one of the most cytokinin responsive genes, reaching expression maxima within 10-15 minutes after application of cytokinins. We argued that a high responsiveness of these genes to cytokinin might hinder detection of more subtle changes in cytokinin sensitivity in line with lower expression of the CRE1/AHK4-GFP. When compared to ARR5 and ARR7, ARR16 type-A homologue displays transcription maximum within 40-60 min after cytokinin treatment (D'Agostino et al., 2000). A significantly higher expression of ARR16 15 minutes after cytokinin application was detected in both CRE1/AHK4-GFP overexpressing lines when compared to the wild type.

In support of functionality of the construct, we show that plants overexpressing CRE1/AHK4-GFP exhibit phenotypes typical for plants exposed to cytokinins, such as a shorter primary root, slower root growth rate and decreased lateral root density (Fig. S5f-i). In addition, both lines show similar, hypersensitive-like response to exogenous cytokinin treatment, which again speaks in favor of the correct functionality of both lines (Fig. S5j). Importantly, the phenotypic effects seem to be proportional to the expression levels detected in both independent lines (Fig. S5a-c).

The observed expression patterns were found to be similar in both lines and not dependent on the level of expression in the line (1) vs line (2). Thus, based on comparative studies of the two independent lines with different levels of CRE1/AHK4-GFP expression we assume that overexpression of CRE1/AHK4-GFP does not produce any major artefacts.

5. The finding of change in TCS:LUC sensitivity in protoplasts (Fig. S4a) just indicates that there is an altered sensitivity. If that is because there is too much AHK4 made in that background, then that is a big issue. If there is too much AHK4 (from abnormal overexpression) on the ER, could it be placed on the PM in some cells? Basically, is an unnatural state reached and the output of what is seen, just misplacement of too much AHK4?

Response: We would like to bring to the reviewer's attention that protoplast assays using TCS::LUC reporter were performed with single aim to test an overall activity of the 35S::CRE1/AHK4-GFP translational fusion construct. We agree with the reviewer that based on this assay it is not possible to draw any conclusion whether CRE1/AHK4-GFP receptor mediates signaling from the ER or the PM, which was not the aim of this experiment.

6. The level of iP-NBD used in the TCS-GFP assay Fig. S1 is very low compared to what is used elsewhere in this manuscript. Also the 15h treatment in that assay, again raises issues about breakdown or conjugation.

Response: We thank the reviewer for raising this point. We repeated the experiment and compared the effects of 1 μ M iP-NBD and 1 μ M iP on expression of the pTCSn::ntdTomato:TNOS cytokinin reporter after 6 and 15 hours of treatment (new Fig. S1c). Whereas iP, an active form of cytokinin, significantly increased expression of the cytokinin reporter already 6 hours after treatment when compared to mock control, iP-NBD failed to enhance expression of the reporter. When applied

simultaneously, iP-NBD attenuated iP mediated enhancement of the TCS reporter expression in line with a proposed iP-NBD activity as a partial agonist/competitive antagonist of cytokinin receptors.

7. In Figure S3, please change the white arrows to something different in either (a) or (b-e) to help differentiate these, which are indicating quite different things.

Response: *We thank for the note. Arrows were amended to differentiate subcellular reporters co-localizing with iP-NBD from these where no co-localization could be detected.*

8. There should be some mention of pH and receptor binding somewhere in the text, as this has been raised as an issue before in relation to the acidity of different organelles.

Response: *This is an important point and we agree this is worth to discuss.*

To our knowledge, the issue of pH has been discussed in respect to pH influence on specific binding of CKs to their receptors. This had been then related to their potential subcellular localization. It was shown in Romanov et al. (2006) that CK binding to AHK3 is pH dependent with optimum in basic pH and dramatic decrease in acidic pH. This finding is fitting with the ER-localization of CK receptors and has also been used to doubt PM-localization of the receptors pointing to acidic pH of apoplast as a constrain of efficient CK binding. However, in contrary to AHK3, CRE1/AHK4 affinity was shown to be not dramatically altered in acidic pH (Romanov et al., 2006). Importantly, recent work by Jaworek et al., 2019 shows detailed analysis of pH influence on binding strength of cytokinins to cytokinin receptors of poplar. They showed that cytokinin binding to PchK3 (ortholog of AHK3) steadily increases towards higher pH values, whereas binding to PchK4 (ortholog of CRE1/AHK4) linearly decreased from a pH optimum for ligand binding at 5.5. These evidences are supporting the idea that CRE1/AHK4 can effectively sense CKs from the apoplast where pH is at 5.0 to 6.0.

Moreover, newly we show here that iP-NBD is pH stable (Fig. S2a). The stability of iP-NBD in a broad range of pH was already mentioned above.

9. What happens in the other basic cell types in the root? There was a lot of imaging in epidermal and LRC cells, but what about other cell types that are very nearby in the root? It would be nice to indicate if there is a general status of mostly being ER localized in the root, as had been seen before in leaf cells, and the epidermis is the exception, or does every cell type need a serious investigation to determine this and it may be very cell type dependent?

Response: *Our observations indicated that in differentiated cells of the lateral root cap (LRC) the CRE1/AHK4-GFP localizes primarily at the ER, whereas in epidermal cells of the root apical meristem cytokinin receptor was detected at the ER, the PM and at the cell plate of dividing meristematic cells.*

In the revised manuscript, we extended our analyses and monitored the CRE1/AHK4-GFP localization in several additional cell types. We show that similarly to epidermis also in provascular cells at the root meristematic zone the CRE1/AHK4-GFP localizes at the ER, the PM and at the cell plate of dividing stele cells (Fig. S6g). To strengthen further conclusion that in meristematically active cells cytokinin receptor might enter secretory pathway and to reach the PM, we performed real time monitoring of the CRE1/AHK4-GFP in developing lateral root primordia (LRP). Although expression of CRE1/AHK4-GFP driven by 35S was weaker in LRP, similarly to cells in the root meristem LRPs the CRE1/AHK4-GFP tend to localize at the ER and the PM. Furthermore, in actively dividing cells we

could detect a weak CRE1/AHK4-GFP signal during cell plate formation (Fig. S6h, Supplemental movie 1).

Unlike cells located at the root apical meristem, in the differentiated cells of the LRC the CRE1/AHK4-GFP was detected in the ER, but not at the PM. To support further our conclusion about dominating localization of the cytokinin receptor at the ER in differentiated cells we performed thorough observation of the CRE1/AHK4-GFP in differentiated epidermal cells of root elongation zone. As expected in these cells the CRE1/AHK4-GFP was located at the ER, but no co-localization with the PM could be detected (Fig. S6i-l).

Reviewer #2 (Remarks to the Author):

The authors use two independent approaches to address a long standing question of the field: what are the cellular compartments from which cytokinin signaling is initiated? Signaling from the ER has been postulated based on the observations that the majority of receptors seems to localize to the ER membranes, with their sensing domain facing the lumen. However, receptors have also been observed at the PM, and based on recent work with the cytokinin transporter PUP14 the apoplast was proposed to be important for signal initiation.

First, the authors use a fluorescently labeled cytokinin (iP-NBD) as a tool to study the subcellular localization of cytokinin-receptor complexes in different cell types.

The attachment of the fluorochrome to the cytokinin iP reduces the affinity between ligand and AHK3, or AHK4 receptors about 100 fold. Also, the modified ligand is unable to initiate a signaling response. Despite these limitations, the authors could show by competition assays, that (iP-NBD) binding is specific and therefore, the fluorescence emitted by is indicative of cytokinin subcellular localization.

In addition, they study the subcellular localisation of the AHK4 cytokinin receptor, translationally fused to GFP. Importantly, they could show the functionality of the used transgene as it was functional in planta in absence the endogenous AHK4.

Extensive studies with different treatments and co-stainments with different cellular markers were performed and revealed that receptors and receptor-ligand complexes are found in the endoplasmic reticulum. In epidermal cells of the root meristem, receptors and receptor-ligand complexes were additionally shown to enter the endomembrane trafficking system. The AHK4-GFP protein was also detected at the plasma membrane and at newly forming cell plates, supporting the possibility that AHK4 at the PM may be involved in signaling initiation. Together with the data presented in the parallel submitted manuscript, the existence of cytokinin-signaling initiation from the PM, in particular mediated by AHK4, is convincingly established.

This work presented here is of high quality work at all levels. Keep in mind that the questions addressed are technically very difficult, as cytokinin ligands and cognate receptors are basically impossible to detect in situ. Therefore, the authors need to use somewhat imperfect tools: iP-NBD has lower affinity to receptors and does not initiate signaling, 35S::AHK4-GFP does show some overexpression phenotypes. However, with all controls and given the fact that both approaches independently provide similar results, the overall conclusions are well supported.

I have only one question, or suggestion: I would expect iP-NBD staining in an *ahk4* and/or *ahk3/4* mutant could be a good negative control, and it would be helpful to perform and present that data.

Response: We agree that the proposed experiment would help to support conclusion about iP-NBD affinity to the cytokinin receptor. However, in our assay we detected only modest differences in the iP-NBD signal in tested ahk4 and ahk3,ahk4 mutant alleles. Overall, we find interpretation of these experiments difficult as, we cannot exclude that although with lower affinity, in addition to CRE1/AHK4, iP-NBD might bind also to other cytokinin receptors compensating for loss of CRE1/AHK4 function.

Reviewer #3 (Remarks to the Author):

In this manuscript, the authors developed a fluorescently labeled cytokinin probe (iP-NBD), and analyzed subcellular localization of the probe and GFP-fused AHKs, specially focusing on CRE1/AHK4. They found action of the compound on the receptor as an antagonist having higher affinity to CRE/AHK4 than other receptors. Then, the authors carefully analyzed subcellular localization of the fluorescent probe and CRE1/AHK4-GFP in lateral root cap cell and epidermal cell at root meristematic zone. Based on the observed results, they suggested that CRE1/AHK4 can enter secretory pathway via Golgi body and reach the plasma membrane. They also observed localization of iP-NBD, and suggested cytokinin could be perceived both ER and plasma membrane. This study provides some important information for cytokinin perception at plasma membrane and ER. Especially, several important findings were shown for deeper understanding of intracellular traffic of cytokinin receptor. On the other hand, most of the data for “functionality” of the receptors are indirect. I raised several points to be addressed.

1: Title is not attractive, but just descriptive.

Response: We changed the title to “Cytokinin fluoroprobe reveals multiple sites of cytokinin perception at plasma membrane and endoplasmic reticulum”

2: The authors used benzyladenine, an artificial cytokinin, to show antagonistic effect of iP-NBD (Fig. 1d). I strongly suggest the authors to use authentic cytokinins, such as iP. In other experiments, they used authentic ones. I would like to know why they used BA in this experiment.

Response: Thank you for the note. We used BA as an active form of cytokinin, which exhibits a good stability in biological assays, however we agree with the criticism that in the cases when comparisons of iP-NBD vs natural CKs are made, BA might not be the most suitable positive control. The relevant experiments (newly Fig. 1c, d; S1a, b, c; S2e) were repeated using iP as an active form of cytokinin. These experiments are described above in response to questions raised by the first reviewer.

3: Line 73, the authors described K_i values of iP-NBD for AHK3 (~37 μ M) and CRE1/AHK4 (~1.4 μ M). But it is difficult to see the values from Fig 1c, especially AHK3 + iP-NBD.

Response: The IC_{50} values were experimentally determined by competition binding assays. The K_i values were then calculated using equation $K_i = IC_{50}/(1 + [radioligand]/K_D)$ by Cheng and Prusoff (1973). The two numbers thus cannot be identical and K_i cannot be directly visible from the graph as IC_{50} does. Calculation of K_i values is recommended because in contrast to IC_{50} , K_i is taking into account concentration of the radioligand used in the assay and thus making this constant transferrable among independent measurements.

4: Information of iP and iP-NBD concentrations is missing in Fig. 1e.

Response: We used standard iP-NBD concentration adopted for staining experiments throughout the manuscript (5 μ M) and the same concentration of the parental cytokinin, iP.

5: The subcellular localization pattern of AHK-GFP fluorescence was relatively clear, whereas that of iP-NBD fluorescence was not. The fluorescence appears to be uniformly distributed. To show the specific binding, they showed Ade-NBD in lateral root cap cells as a control in Fig. S1g, but more information, such as the data of epidermal cells should be provided. Therefore, it is not sure whether the overlap with the subcellular localization pattern of other marker proteins is specific event or not.

Response: We agree that co-localization of iP-NBD with some marker lines and quality of images was not optimal. In the revised manuscript, we repeated the iP-NBD localization studies using improved imaging system with super-resolution detection mode (ZEISS Airyscan 2 detector) installed recently in our bio-imaging facility. We believe that the quality of the iP-NBD localization has improved, and clearly supports our conclusions about affinity of iP-NBD to subcellular compartments in LRC (Fig. 2a,e,f; S2a,S3) and meristematic epidermal cells (Fig. 2b,c,d,g; S3). In addition, we quantified co-localization of iP-NBD with genetically-encoded molecular markers for diverse subcellular compartments using Pearson correlation coefficient as suggested by the reviewer 4 (Table 1). As concerns Ade-NBD localization, we show in both cells of the LRC and epidermal cells that the accumulation in cells is very low when compared to iP-NBD and even at increased contrast, they do not exhibit subcellular localization as observed for iP-NBD (Fig. S2f), again pointing to the specificity of iP-NBD interaction with AHK receptor.

Reviewer #4 (Remarks to the Author):

Cytokinins are essential plant growth phytohormones which promote plant cell growth, division and differentiation as well as various developmental and physiological processes. In this study, Kubiasová et al., have developed a useful cytokinin fluorescent probe termed iP-NBD with higher affinity to CRE1/AHK4 cytokinin receptor. By using the probe, the authors further demonstrated that the cytokinin receptor CER1/AHK4 participates in the endomembrane trafficking system from ER to the plasma membrane, possibly involved in mediating cytokinin signaling and gradient in plant roots. In contrast to the conventional ER-localization of cytokinin receptor, demonstration of the new subcellular localization of CER1/AHK4 on the PM provides a new perspective of cytokinin signaling pathway. The findings of the study are novel and interesting to the field. In addition, it may help to understand of establishment of cytokinin response gradient in different plant organs since the detailed

underlying mechanisms are not well understood. However, I would suggest the authors to address the following concerns:

1. The image quality in some of the Figures and Supplementary Figures is poor. The real fluorescent signal is difficult to be recognized in the high noise background. Meanwhile, such images are neither representative nor convincing enough to support the conclusions as stated in the manuscript. For example, the iP-NBD background noise in Fig. 2b is too high to separate the real signal and noise. It is obviously not convincing enough to demonstrate that iP-NBD localizes in ER although ER specific

marker was used for the colocalization study. The ER marker showed the typical ER patterns but the iP-NBD did not. Similar problems also can be found in Fig. S3b, c and d.

Response: We agree with the reviewer, that subcellular localization of iP-NBD with some genetically encoded molecular markers for diverse subcellular compartments and quality of images was not optimal. In the revised manuscript, we repeated the iP-NBD localization studies using improved imaging system with super-resolution mode (Airyscan detector, installed recently in our bioimaging facility). We believe that the quality of the iP-NBD detection has improved, and together with quantification of co-localization using Pearson correlation coefficient (Table 1), our conclusions about affinity of iP-NBD to subcellular compartments in LRC (Fig. 2a,e,f; S3) and meristematic epidermal cells (Fig. 2b,c,d,g; S3) are justified.

2. For all of the colocalization and non-colocalization studies in all of the Figures, the authors are supposed to perform quantitative and statistic analysis to clearly demonstrate the ratio of the colocalization. It is especially useful for illustration of the cases of partial colocalization and non-colocalization of two different organelles/vesicles, such as in Fig. 2, Fig. S2a, Fig.3 and Fig. S3.

Response: As mentioned above, we repeated most of the experiments using improved imaging system in super-resolution mode and included quantification of Pearson correlation coefficient (Table 1).

3. The time of FM4-64 endocytosis should be clarified in the text and figure legend since the FM4-64 is firstly endocytosed to the early endosome (TGN) and then goes to the late endosome (MVB), and if given enough time, it will eventually reach the tonoplast of the vacuole. What are the FM4-64 cytosolic dots in Fig 2 mainly representing for?

Response: Thank you for pointing this out. Co-localization of iP-NBD with FM4-64 was observed within 15 minutes, indicating localization in the early endosomes/TGN. The missing information was added to figure legend (Fig. S3c).

4. The conclusion stated in the text line 133-136 can not be fully supported by Fig.2 c,e and Fig. S2b. The central vacuole in plant cells is usually large and it pushes the other organelles and cytosol to the cortical areas which are underneath the PM. Therefore, how to rule out the possibility that the PM pattern highlighted by iP-NBD is actually from ER because iP-NBD in the ER is pushed very close to the PM by the central vacuole, as shown in Fig. 2c and e? Additionally, the florescent intensity analysis as shown in Fig. 2d and f can not support the PM-localization of iP-NBD in plant cells.

Response: Our repeated experiments with improved super-resolution imaging set-up (including Airyscan 2 detector) support our previous conclusion on the partial iP-NBD localization at the PM in the epidermal cells of the root meristem (Fig. 2c,d). On other hand, we would like to note that unlike epidermal cells at the root apical meristem, in LRC cells no co-localization of iP-NBD with the PM marker could be detected as supported by profiling of iP-NBD and PM marker fluorescence intensity distribution (Fig. 2e,f).

5. The authors chose two types of plant cells: LRC and epidermal cell as the studying materials. What are the reasons for selecting them?

Response: The main reason to focus on both cell types was our preliminary observation that subcellular localization of the CRE1/AHK4-GFP might differ in differentiated cells of LRC when compared to epidermal cells at the meristematic zone of the root tip. Whereas in cells of LRC the CRE1/AHK4-GFP located primarily at the ER, in meristematic epidermal cells we noticed that CRE1/AHK4-GFP might also be located at the PM and cell plate in addition to the ER.

As suggested by reviewer 1, in the revised manuscript we extended our analyses and monitored the CRE1/AHK4-GFP localization in several additional cell types.

We show that similarly to meristematic epidermis also in provascular cells at root meristematic zone the CRE1/AHK4-GFP localizes at the ER, the PM and at the cell plate of dividing stele cells (Fig. S6g).

To strengthen further conclusion that in meristematically active cells cytokinin receptor might enter secretory pathway and to reach the PM, we performed real time monitoring of the CRE1/AHK4-GFP in developing lateral root primordia (LRP). Although expression of CRE1/AHK4-GFP driven by 35S was weaker in LRP, similarly to cells in the root primary meristem the CRE1/AHK4-GFP tends to localize at the ER and the PM. Furthermore, in actively dividing LRP cells we could detect a weak CRE1/AHK4-GFP signal during cell plate formation (Fig. S6h, Supplemental movie 1).

Unlike cells located in the root apical meristem and in LRP, differentiated cells of the LRC showed CRE1/AHK4-GFP localization at the ER, but not at the PM. To support further our conclusion about dominating localization of the cytokinin receptor at the ER in differentiated cells, we performed thorough observation of the CRE1/AHK4-GFP in differentiated root epidermal cells in root elongation zone (localized above the meristematic zone). In these cells the AHK4-GFP was located at the ER, but no co-localization with the PM could be detected (Fig. S6i-l).

6. The subcellular localization pattern of cytokinin receptor CRE1/AHK4 in two different plant cell types are different (one is mainly on ER and the other one on both ER and PM), however, there is no further discussion to illustrate the possible reasons. Except for CRE1/AHK4, how about the other different cytokinin receptors (also localized to PM or depending on different cell types)?

Response: We cannot exclude that other receptors can also be located at the PM. In this context it is worth mentioning that CRE1/AHK4 is mainly expressed in the root, it can be found in the vascular cylinder and the pericycle cells (Higuchi et al., 2004). In contrast, AHK3 expression is more associated with the shoot of Arabidopsis plants (Ueguchi et al., 2001; Higuchi et al., 2004).

*In previously published studies localizations of the AHK3-GFP and AHK2-GFP have been observed in above-ground plant parts using transiently transformed *Nicotiana benthamiana* epidermal leaf cells (Wulfetange et al., 2011) and transiently transformed *Arabidopsis* cotyledon cells (Caesar et al., 2011), all in the differentiated stages. Hence, studies that are more detailed would be needed to examine whether similarly to the CRE1/AHK4-GFP in meristematically active cells, also AHK2 and AHK3 might enter secretory pathway and reach the PM. As it concerns signaling downstream of the CRE1/AHK4 receptor located at either the ER or the PM this is an intriguing question. At this moment, we can only speculate about distinct pathways activated by the CRE1/AHK4-GFP located at either of these compartments to control specific process in differentiated vs meristematically active cells. Detailed dissection of these pathways would require further investigations. As suggested by functional analyses of the PUP14 cytokinin transporter, there might be pool of cytokinins located in the apoplast that should be sensed by membrane located cytokinin receptor.*

7. Cytokinin mainly consists of two active types: iP and tZ. In this study, the author use iP-NBD as a cytokinin reporter. What about tZ? Do the two forms of cytokinin show the similar or different subcellular localization patterns? Compared to the results obtained from Antoniadin et al., how to

explain and make sense the differences in functional roles and responses of iP and tZ in the two independent studies?

Response: Thank you for addressing these important points. Indeed, tZ and iP do show different localization patterns in plants. Generally, it seems that tZ-type cytokinins play a role of acropetal messengers, whereas iP-type cytokinins operate as systemic or basipetal messengers (Kudo et al., 2010). In line with this, detailed analysis of all major active cytokinin forms (iP-, tZ- and cZ-types) showed a presence of a cytokinin gradient within the root tip of Arabidopsis – the concentration peak was detected in LRC cells, columella, columella initials, and QC cells (Antoniadi et al., 2015). This CK concentration distribution pattern roughly matches the observed expression pattern of CRE1/AHK4-GFP in both transgenic lines. In addition, Antoniadi et al. also previously showed that cZ-type cytokinins were the predominant forms in Arabidopsis root apices followed by iP types, while tZ-type CKs were found to be less abundant (5-fold lower concentrations), however, the CK distribution pattern of isoprenoid CKs (tZ, cZ, iP-types) showed similar distribution patterns in different cell type populations within the root apex. In contrast, when concerns free CK bases the tZ content was found to be the highest among the free bases, whereas free iP showed relatively enhanced content also in the stele. For these reasons, iP seems to be a good candidate for a cytokinin fluoroprobe design, also bearing in mind that its side chain does not contain a free hydroxyl group (that can be modified by O-glycosylation) as previously mentioned.

As described in the manuscript the selection of the cytokinin fluoroprobe was also done with intention to minimize the possibility of metabolic conversion by cytokinin deactivation enzymes in planta. For this reason isopentenyladenine (iP) was used as a natural cytokinin that cannot be transformed through O-glycosylation at the cytokinin side chain. From the technical point of view, the preparation of tZ accompanied with fluorescent label proved to be rather challenging due to the presence of the aforementioned hydroxyl functional group in C6-substituted isoprenoid side chain and would deserve independent project. Hydroxyl would have to be protected by protective group, which would have been unprotected (usually under harsh conditions) after the linker addition and fluorescent label attachment.

In their current work, Antoniadi et al. show that iP and tZ (although only tZ was found to be significantly enriched in TCSn::GFP⁺ cells) both triggered cytokinin responses in the TCS protoplast assay when supplied exogenously, i.e. in the form of conjugates covalently attached to Sepharose beads. Other lines of evidence suggest that iP and tZ have distinct biological roles. Importantly, using TCSn::GFP they show that iP had the strongest effect on meristematic stele cell initials, whereas tZ response was maximal in the transition zone in the stele. Interestingly, the TCS signal in those cells was absent in ahk4 mutant line.

We believe this evidence complements our data and points to tissue-specific roles of iP and tZ that needs to be correlated with PM vs ER spatially heterogeneous distribution of cytokinin HK receptors in different cell populations of the root meristematic and differentiation zones.

Reviewers' comments:

Reviewer #1 (Remarks to the Author):

The authors have spend considerable effort in addressing both my and other reviewers comments. Overall I am satisfied with the revised version of the manuscript. Nearly all of my many questions/comments have been answered and appear to support the authors findings as written in the manuscript. Using iP as a control and the modeling of the iP-NBD into the receptor pockets along with competitive binding assays strengthens the argument being made. Discussion of CXK2 as apoplastic and largely relevant for exogenous CK application are good additions. The pH stability is also quite nice to see. Results regarding the 35S:AHK4 line are good to see, and it would see that more work on this could be its own spinoff paper. New microscopy is nice and even though the main point is about receptor placement to the PM, the images that place it on the ER are very clear. The few minor point that I still have about things, I feel that the authors have convinced me that they are not directly impacting on the points being directly addressed in this study, thus do not need more addressing.

Reviewer #3 (Remarks to the Author):

In the revised version, the authors have appropriately addressed all points I raised. Quality of images has been improved sufficiently.

Reviewer #4 (Remarks to the Author):

The authors have already addressed most of my concerns. Nevertheless, for better clarification, it will be beneficial to include the responses of point 6 and 7 as a part of Discussion in the manuscript since readers in the field could have similar concerns and questions. I would like to recommend it for publication in Nature Communications after the discussion is further improved as mentioned above.

Reviewer #5 (Remarks to the Author):

This is interesting work and elegantly demonstrated the perception of CK by AHK receptors at a subcellular level. In the view of the bioorganic chemist, I still have concern for the subcellular localization of iP-NBD.

In this study, iP-NBD was shown as a weak agonist/ partial antagonist of AHK4. Fig. 1e represented that the iP reduced the iP-NBD accumulation in cells and Fig. 1f showed that NBD signal was reduced in pup14 mutant. As mentioned on line 146 "This suggested that transport and/or intracellular binding competition between iP-NBD and the natural cytokinin competitor was taking place", thus, without further evidence, these data could not support the conclusion that iP-NBD can compete with iP at AHK receptor in planta. Fig. 2 and other pictures displayed that the subcellular distribution of iP-NBD. This is very impressive and would be firm evidence of the subcellular distribution of iP-NBD molecules at PM and ER.

I have a question on an important issue. This subcellular distribution of iP-NBD really depends on AHK CK receptors? The iP-NBD signals are result from the AHK bound signals or free iP-NBD or both bound/free iP-NBD? If the fluorescent signals are derived from both free/bound form, how is the ratio of bound/free form? I think that this is a critical issue. If the most of subcellular iP-NBD signal is resulting from a free iP-NBD molecule, such image would not indicate the site of CK perception, but only demonstrate the subcellular localization of "synthetic chemicals" that transported and accumulated by unidentified proteins (pathways). Thus, the experimental approach using iP-NBD might not support the conclusion in this work. The binding of iP-NBD to CK

receptors "in planta" would be essential for the subcellular distribution of iP-NBD in PM and ER. The validation for the in planta binding of iP-NBD to receptors would be essential data in this manuscript. This is also pointed out by another reviewer. On the crucial experiment using ahk3/4 mutant, authors failed to show clear results due to the redundant function of receptors and weak affinity of iP-NBD on AHKs.

In the response sheets,

>iP-NBD staining in an ahk4 and/or ahk3/4 mutant could be a good negative control, and it would be helpful to perform and present that data.

Response: We agree that the proposed experiment would help to support conclusion about iP-NBD affinity to the cytokinin receptor. However, in our assay we detected only modest differences in the iP-NBD signal in tested ahk4 and ahk3,ahk4 mutant alleles. Overall, we find interpretation of these experiments difficult as, we cannot exclude that although with lower affinity, in addition to CRE1/AHK4, iP-NBD might bind also to other cytokinin receptors compensating for loss of CRE1/AHK4 function.

To reveal this issue, I would like to propose the following experiments. Authors already performed similar experiment in Fig. 1. However, this was carried out for the uptake experiment of iP-NBD and the subcellular distribution image of iP-NBD have not analyzed.

The subcellular distribution image of iP-NBD can be examined in the presence of the excess active CKs iP, tZ, benzyladenine, kinetin and urea-type CK agonists (CPPU and Thidiazuron) and negative control, adenine and N,N-dimethyladenine (sigma D2629-100MG) by adding each chemical and combined treatment (to overcome the selectivity of agonist on distinct receptors). If iP-NBD distribution depends on the AHK binding as expected, the NBD signal would disappear like negative control adenine-NBD? or the spot-like subcellular structures would be changed in the image?

The negative control Ade-NBD might be nice one, additionally, N,N-dimethyl adenine-NBD used as negative control in author's previous work (Kubiasová et al, phytochemistry 150, 2018, 1-11) should be included in these in planta competitive assay of iP-NBD imaging. Based on the working model in this study, the image of iP-NBD would be affected by the competition with active CK and CK agonist, but the image of Ad-NBD and N,N-dimethyl-Ad-NBD would not be affected by both active CK and inactive control. When the excess active CK was added after iP-NBD treatment, the signal intensity of iP-NBD would be decreased?

If the ratio of bound/free iP-NBD signal in the image is considerably low value, the difference in the images after the competition with CK might be subtle. In such case, it might be technically challenging to see the difference between bound/free statuses of iP-NBD on AHK in planta. In my view, the evidences by the competition experiments above mentioned would convincingly support the conclusion that the subcellular distribution of iP-NBD displays the site of CK perception.

We would like to thank all reviewers for insightful and helpful comments. We were pleased to see that all previous reviewers were satisfied with revision of our manuscript.

Reviewers' comments:

Reviewer #1 (Remarks to the Author):

The authors have spend considerable effort in addressing both my and other reviewers comments. Overall I am satisfied with the revised version of the manuscript. Nearly all of my many questions/comments have been answered and appear to support the authors findings as written in the manuscript. Using iP as a control and the modeling of the iP-NBD into the receptor pockets along with competitive binding assays strengthens the argument being made. Discussion of CXX2 as apoplatic and largely relevant for exogenous CK application are good additions. The pH stability is also quite nice to see. Results regarding the 35S:AHK4 line are good to see, and it would see that more work on this could be its own spinoff paper. New microscopy is nice and even though the main point is about receptor placement to the PM, the images that place it on the ER are very clear. The few minor point that I still have about things, I feel that the authors have convinced me that they are not directly impacting on the points being directly addressed in this study, thus do not need more addressing.

Reviewer #3 (Remarks to the Author):

In the revised version, the authors have appropriately addressed all points I raised. Quality of images has been improved sufficiently.

Reviewer #4 (Remarks to the Author):

The authors have already addressed most of my concerns. Nevertheless, for better clarification, it will be beneficial to include the responses of point 6 and 7 as a part of Discussion in the manuscript since readers in the field could have similar concerns and questions. I would like to recommend it for publication in Nature Communications after the discussion is further improved as mentioned above

Response: We thank the reviewer for the suggestion. We extended the discussion related to these points in the revised manuscript.

Reviewer #5 (Remarks to the Author):

This is interesting work and elegantly demonstrated the perception of CK by AHK receptors at a subcellular level. In the view of the bioorganic chemist, I still have concern for the subcellular localization of iP-NBD.

In this study, iP-NBD was shown as a weak agonist/ partial antagonist of AHK4. Fig. 1e represented that the iP reduced the iP-NBD accumulation in cells and Fig. 1f showed that NBD signal was reduced in pup14 mutant. As mentioned on line 146 “This suggested that transport and/or intracellular binding competition between iP-NBD and the natural cytokinin competitor was taking place”, thus, without further evidence, these data could not support the conclusion that iP-NBD can compete with iP at AHK receptor in planta. Fig. 2 and other pictures displayed that the subcellular distribution of iP-NBD. This is very impressive and would be firm evidence of the subcellular distribution of iP-NBD molecules at PM and ER.

I have a question on an important issue. This subcellular distribution of iP-NBD really depends on AHK CK receptors? The iP-NBD signals are result from the AHK bound signals or free iP-NBD or both bound/free iP-NBD? If the fluorescent signals are derived from both free/bound form, how is the ratio of bound/free form? I think that this is a critical issue. If the most of subcellular iP-NBD signal is resulting from a free iP-NBD molecule, such image would not indicate the site of CK perception, but only demonstrate the subcellular localization of “synthetic chemicals” that transported and accumulated by unidentified proteins (pathways). Thus, the experimental approach using iP-NBD might not support the conclusion in this work. The binding of iP-NBD to CK receptors “in planta” would be essential for the subcellular distribution of iP-NBD in PM and ER. The validation for the in planta binding of iP-NBD to receptors would be essential data in this manuscript. This is also pointed out by another reviewer. On the crucial experiment using ahk3/4 mutant, authors failed to show clear results due to the redundant function of receptors and weak affinity of iP-NBD on AHKs.

1. In the response sheets,
>iP-NBD staining in an *ahk4* and/or *ahk3/4* mutant could be a good negative control, and it would be helpful to perform and present that data.

Response: We agree that the proposed experiment would help to support conclusion about iP-NBD affinity to the cytokinin receptor. However, in our assay we detected only modest differences in the iP-NBD signal in tested *ahk4* and *ahk3,ahk4* mutant alleles. Overall, we find interpretation of these experiments difficult as, we cannot exclude that although with lower affinity, in addition to CRE1/AHK4, iP-NBD might bind also to other cytokinin receptors compensating for loss of CRE1/AHK4 function.

Extended comment: Our conclusion is also supported by molecular docking analysis, which suggests possibility of iP-NBD binding into all three AHK isoforms (Fig. 1b). The ability of high excess of iP-NBD to compete the binding of tZ to AHK3 is presented in the Fig. S1a.

To reveal this issue, I would like to propose the following experiments. Authors already performed similar experiment in Fig. 1. However, this was carried out for the uptake experiment of iP-NBD and the subcellular distribution image of iP-NBD have not analyzed.

The subcellular distribution image of iP-NBD can be examined in the presence of the excess active CKs iP, tZ, benzyladenine, kinetin and urea-type CK agonists (CPPU and Thidiazuron) and negative control, adenine and N,N-dimethyladenine (sigma D2629-100MG) by adding each chemical and combined treatment (to overcome the selectivity of agonist on distinct receptors). If iP-NBD distribution depends on the AHK binding as expected, the NBD signal would disappear like negative control adenine-NBD? or the spot-like subcellular structures would be changed in the image?

The negative control Ade-NBD might be nice one, additionally, N,N-dimethyl adenine-NBD used as negative control in author's previous work (Kubiasová et al, phytochemistry 150, 2018, 1-11) should be included in these in planta competitive assay of iP-NBD imaging. Based on the working model in this study, the image of iP-NBD would be affected by the competition with active CK and CK agonist, but the image of Ad-NBD and N,N-dimethyl-Ad-NBD would not be affected by both active CK and inactive control. When the excess active CK was added after iP-NBD treatment, the signal intensity of iP-NBD would be decreased?

If the ratio of bound/free iP-NBD signal in the image is considerably low value, the difference in the images after the competition with CK might be subtle. In such case, it might be technically challenging to see the difference between bound/free statuses of iP-NBD on AHK in planta. In my view, the evidences by the competition experiments above mentioned would convincingly support the conclusion that the subcellular distribution of iP-NBD displays the site of CK perception.

Response: We thank the reviewer for insightful comments. All raised points are relevant and we agree that some might not be addressed exhaustively in our manuscript.

1. We agree that results of transport assays presented at Fig.1e cannot be understood as an evidence for the iP-NBD binding to the AHK receptors. The main objective of this experiment was to test cytokinin-like properties of iP-NBD including its transport to cell. However, we believe that reduced signal of iP-NBD in cells when co-applied with iP might not only result from competition between non- and labeled cytokinin for transport, but also binding to the receptor. Therefore when interpreting our results we considered that *both "transport and/or intracellular binding competition between iP-NBD and the natural cytokinin competitor was taking place"*.
2. As it concerns the point whether the binding of iP-NBD is dependent on the AHK receptors and whether iP-NBD signals are result from AHK bound signals or free iP-NBD or both bound/free iP-NBD. This is an interesting question, but not so trivial to address experimentally even using fluorescently labeled cytokinins.

We cannot exclude that part of iP-NBD signal in the cell is not bound, and this might also not be the case for natural endogenous cytokinins. The subcellular localization and sequestration of cytokinin in cell is very complex and so far not resolved question; as extensively discussed in the recent review by Romanov, Lomin and Schmulling (2018). To evaluate ratio between bound and free iP-NBD in cells is technically extremely challenging. The fact that iP-NBD exhibits co-localisation with ER and endosomes makes visualization and reliable quantification of cytosolic signal even more difficult. We would not be able to dissect reliably signal of iP-NBD in cytosol from this associated with ER or endosomes, which is needed to properly evaluate ratio between free and bound iP-NBD. Saying that implicates difficulties for detecting potential differences in subcellular pattern of iP-NBD in cytokinin receptor mutants. If, according to expectations, lack of the receptor results in reduced iP-NBD signal

associated with ER and endosomes and non-bound iP-NBD would accumulate in cytosol, we face the challenge of reliable quantification. In addition, it has to be considered that in single cytokinin receptor mutants besides compensation by other homologues receptors, there might be feedback on other components of pathways contributing to cytokinin homeostasis including transport, degradation, or conjugation. These are the main reasons why we find experiments and interpretation of such results difficult.

Again, we understand the point of the reviewer and we even tried to address it by analyzing iP-NBD pattern in cytokinin receptor mutants. However, we have to admit that to perform reliable bioimage analyses that would inform us whether in cytokinin receptor mutants there is less iP-NBD co-localisation with ER, endosomes and higher accumulation in cytosol, which would lead to shift in ratio between free and bound iP-NBD when compared to wild-type is almost impossible. We can share with the reviewer results from analyses of the *ahk2*, *ahk3*, *ahk4* triple mutant where we detected reduced accumulation of iP-NBD in root epidermal cells when compared to wild type. However, the mutant lacking all three receptors exhibits severe developmental defects, seedlings are small with short roots. Thus, reduced iP-NBD signal in cells might be result of less efficient transport or overall affected fitness of these seedlings (Revision Fig.1).

Revision Fig. 1: Localization of iP-NBD (5 μ M) in *Arabidopsis* epidermal cells stained for 8 min in wild type (Col-0) and triple *ahk* mutant. All pictures were taken with the same settings. Col-0 without any treatment was used as a control (A). Graph shows kinetics of 5 μ M iP-NBD uptake in root epidermal cells of wild type and triple *ahk* mutant. Fluorescence was measured in 3 time points (0, 4, 8 min). The bars represent average \pm s.e., **** = $p < 0.0001$; $n \geq 30$ (Student's *t*-test) (B). Scale bar = 5 μ m.

In this context, we believe that also experiments proposed by the reviewer to perform co-treatment of iP-NBD with cytokinin analogues in wild-type and cytokinin receptor mutant background would suffer from the same technical problems - reliable quantifications of iP-NBD subcellular localization in ER, endosomes and cytosol and potential differences in wild-type and mutant background. In addition, we believe, in agreement with other reviewers' view, that iP is the proper control for iP-NBD-based experiments, also bearing in mind that iP-type cytokinins represents one of the major CK groups in the root apex that is significantly more abundant than tZ-type cytokinins (Antoniadi et al. 2015). Including different cytokinin isoforms might further complicate interpretation of results as various cytokinins might not only competing for binding the AHK receptors differently, but also enter cells with distinct kinetics. Using Ade-NBD as a negative control is certainly a good suggestion, we have already performed series of bioimaging experiments, but due to the low intracellular signal of this derivative

(possibly resulting from a weaker membrane transport) we unfortunately face the same limitation – unreliable and insignificant quantification of results.

- We have to object to the criticism related to lack of support for iP-NBD binding cytokinin receptor *in planta*. As explained above, it is difficult to provide evidence by classical cell bioimaging approaches, however analyses using cytokinin sensitive reporters such as *ARR5::GUS* and *pTCSn::ntdTomato::TNOS* clearly show that iP-NBD interferes with cytokinin signaling and might acts as cytokinin agonist in planta (Fig 1d and Fig S1c). In addition, we would also like to share with the reviewer results we presented in the first version of our manuscript demonstrating that iP-NBD antagonizes cytokinin mediated stimulation of *ARR5* reporter in wild-type and *ahk2,ahk3* mutant, in which *AHK4* receptor is still functional. On the other hand, in *ahk2,ahk4* mutant lacking *AHK4* receptor antagonizing effect of iP-NBD on cytokinin induced expression of *ARR5* was significantly less pronounced. We believe that also these experiments support iP-NBD interaction with *AHK4* receptor mediated cytokinin signaling (Revision Fig.2).

Revision Fig. 2: Quantitative evaluation of β -glucuronidase activity in *Col-0* and *ahk* double receptor mutants harbouring *ARR5::GUS* after incubation with cytokinin *N*6-benzyladenine (BA), iP-NBD and their combination. Optimal concentration to reach the highest reporter response used was 1 μ M. The concentration of iP-NBD was 10 μ M, mock treatment represents solvent control DMSO (0.1%). The bars represent mean \pm s.d., $p < 0.01$ by ANOVA test, ($n=3$). MU, 4-methylumbelliferone.

As originally the experiments were performed using *N*⁶-benzyladenine (BA) we recommend to repeat these experiments with iP as the most relevant cytokinin analogue. Unfortunately, lines carrying *ARR5* reporter in cytokinin mutant background lost germination capacity and therefore we were able to perform these experiments only using wild-type control (Fig 1d of the current manuscript). We found that similarly to previous observation using BA, cytokinin derivative iP-NBD interfered with iP mediated induction of *ARR5* expression.

References:

- Antoniadi, I. et al. Cell-type-specific cytokinin distribution within the Arabidopsis primary root apex. *Plant Cell* 27, 1955-1967 (2015).
- Romanov, G. A., Lomin, S. N. & Schmülling, T. Cytokinin signaling: from the ER or from the PM? That is the question! *New Phytol.* 218, 41-53 (2018).

REVIEWER COMMENTS

Reviewer #5 (Remarks to the Author):

Thank you very much for sharing additional results for my review. On the image analysis of a cell co-treated with iP-NBD and cytokinins, I fully agree with author's opinion. The image analysis would have considerable technical difficulties as author stated. However, I still have a concern on AHK agonist/antagonist activity of iP-NBD in planta. In the present MS, the binding of iP-NBD on AHK receptors in planta was addressed by two reporter assays, ARR5::GUS (16h incubation for assays, Fig 1) and TCSn::Tomato-NLS (16h incubation, Fig S1c).

In previous my review, I mentioned that fluorescent imaging of root cell co-treated with iP-NBD and cytokinins would be convincing data to demonstrate in planta binding of iP-NBD to AHKs, if available. The competition between iP-NBD and iP would be a very short-term event. One of the reasons why I asked the imaging analysis using competition between iP-NBD and cytokinin in planta.

Authors indicated iP-NBD acts as a partial weak agonist based on ARR5-GUS assay (Fig. 1d). However, iP-NBD failed to activate AHKs in bacterial assay (Fig. S1b) implying iP-NBD might be a full antagonist. In ARR5 and TCS reporter assay, the incubation time is so long (16 h) to observe reporter protein expression. Authors also demonstrated that iP-NBD would be drastically metabolized after 1 hour (Fig. S2b). Thus, iP-NBD is continuously supplied into cells from a medium and iP-NBD would be decomposed during a long incubation to release small amount of iP, together with deprenylated Ade-NBD, and the iP released from iP-NBD might activate ARR5::GUS transgene. Because 10 nM iP is sufficiently induce ARR5::GUS same as 10uM iP-NBD (Fig. 1d) (0.1% of 10 uM iP-NBD is equivalent to 10 nM iP). Although it is different systems, bacterial assay indicated that iP-NBD affinity to AHK4 is about 1000 times less active than iP (50% competition values in Fig. S1a) in competition assay, in contrast, 1uM iP-NBD can repress the TCS activation in root by 1 uM iP (molar ratio of iP : iP-NBD= 1:1) in Fig. S1c. The concentration of iP-NBD in this data (Fig. S1c) is correct? If it is correct concentration, how does authors explain these results. iP and iP-NBD are incorporated by same CK transport pathway (Fig. 1e) and iP-NBD is not stable for long time incubation. Thus, iP-NBD would not be accumulated extremely higher than iP. However, iP-NBD can compete with iP at same concentration in root TCS reporter assay. These issues raise the question. iP-NBD really function on AHK in planta ?

If iP-NBD has weak agonist activity in planta, the perception of iP-NBD by AHK occur at PM and ER. This case is simple. The iP-NBD is perceived at PM and ER that iP-NBD is accumulated. On the other hand, If iP-NBD is full antagonist (not agonist), the interpretation would be so complicated. Because the experiment using antagonist (iP-NBD) always involve another molecule, agonist (iP). The transport, metabolic stability, endogenous/exogenous agonist, receptor selectivity, all these issues would affect the inhibition of agonist activity by the antagonist. I certainly know that this would be derived from technical limitations using small-molecule probes.

To address the stability of iP-NBD during the reporter assay and mode of action of iP-NBD in planta, at least, I think the short-term response of plant to iP-NBD is essential data, instead of imaging analysis previously mentioned. Fig. S5e demonstrated that qRT-PCR results of ARR5 gene induction (15 min incubation). In the same way, the agonistic and antagonistic effects of iP-NBD in WT and ahk mutant should be evaluated at the same time-scale (15-40 min, stable condition for iP-NBD) as that used in the fluorescent imaging. It would convincingly support the binding of iP-NBD on AHKs in planta.

SECOND REVISION REVIEWER 5:

Reviewer #5 (Remarks to the Author):

Thank you very much for sharing additional results for my review. On the image analysis of a cell co-treated with iP-NBD and cytokinins, I fully agree with author's opinion. The image analysis would have considerable technical difficulties as author stated. However, I still have a concern on AHK agonist/antagonist activity of iP-NBD in planta. In the present MS, the binding of iP-NBD on AHK receptors in planta was addressed by two reporter assays, ARR5::GUS (16h incubation for assays, Fig 1) and TCSn::Tomato-NLS (16h incubation, Fig S1c).

In previous my review, I mentioned that fluorescent imaging of root cell co-treated with iP-NBD and cytokinins would be convincing data to demonstrate in planta binding of iP-NBD to AHKs, if available. The competition between iP-NBD and iP would be a very short-term event. One of the reasons why I asked the imaging analysis using competition between iP-NBD and cytokinin in planta.

Authors indicated iP-NBD acts as a partial weak agonist based on ARR5-GUS assay (Fig. 1d). However, iP-NBD failed to activate AHKs in bacterial assay (Fig. S1b) implying iP-NBD might be a full antagonist.

In the present MS, the binding of iP-NBD on AHK receptors in planta was addressed by two reporter assays, ARR5::GUS (16h incubation for assays, Fig 1) and TCSn::Tomato-NLS (16h incubation, Fig S1c).

In previous my review, I mentioned that fluorescent imaging of root cell co-treated with iP-NBD and cytokinins would be convincing data to demonstrate in planta binding of iP-NBD to AHKs, if available. The competition between iP-NBD and iP would be a very short-term event. One of the reasons why I asked the imaging analysis using competition between iP-NBD and cytokinin in planta.

Authors indicated iP-NBD acts as a partial weak agonist based on ARR5-GUS assay (Fig. 1d). However, iP-NBD failed to activate AHKs in bacterial assay (Fig. S1b) implying iP-NBD might be a full antagonist.

Response:

First of all, we would like to thank the Reviewer for her/his critical comments. We fully share the interest to gain detailed molecular insights into mechanism through which iP-NBD interacts with cytokinin receptor and we believe that in this response we brought more evidences supporting our assumptions.

The *E. coli* competitive assay (Romanov et al., 2005) clearly shows that iP-NBD is recognized by cytokinin receptors as a ligand, although with app. 1,000-times lower affinity compared to iP (Fig. S1a). The question remained whether the interaction of iP-NBD with receptor can cause its activation. The answer to this question can point to agonistic/antagonistic mode of action. Using the cytokinin response receptor activation assay in *E. coli* ($\Delta rcsC$, *cps::lacZ*) with recombinant AHK3 and CRE1/AHK4 receptors we could not detected significant increase of the cytokinin reporter expression within 16 hours. Taking into account the difference between K_i values of iP and iP-NBD found in the competitive assay (Fig S1a), we decided to prolong incubation time to 24 hours, and in this way to increase threshold for detection of iP-NBD activity. We performed such an experiment with AHK4-expressing *E. coli* and compared the effect of iP-NBD with respective negative controls (Ade, Ade-NBD). As shown in the Revision Figure 1 (below), under the conditions of prolonged incubation we measured a significant activation of the AHK4-mediated signaling pathway by 50 μ M iP-NBD, whereas Ade and

Ade-NBD didn't differ from the mock control. This result supports the previous finding that iP-NBD might act as a weak agonist in the *ARR5:GUS* assay (Fig. 1d).

Revision Figure 1: Effect of iP, iP-NBD and respective negative controls (Ade, Ade-NBD), all in 50 μ M concentration, on activation of AHK4 in *E. coli* activation assay after prolonged incubation (mean \pm s.d; *** $p < 0.001$, ** $p < 0.01$; Student's t-test indicates significant difference when compared to mock treatment, $n = 3$). Left, comparison of all compounds tested; Right, close-up to the comparison of iP-NBD and the respective controls.

Moreover as suggested by the reviewer, we performed more thorough analyses of the iP-NBD activity in plants (Fig 1d). We found that iP-NBD in a concentration dependent manner significantly increases expression of the early cytokinin response gene *ARABIDOPSIS RESPONSE REGULATOR5 (ARR5)* already 15 minutes after its application, suggesting that the synthetic cytokinin fluoroprobe can activate cytokinin signalling pathway *in planta* (Fig. 1d, Fig. S1). In comparison to iP, a natural cytokinin, iP-NBD triggered cytokinin response with significantly lower efficacy and when applied together with iP no additive effect on the *ARR5* expression could be detected (Fig. 1d). In the *pTCSn::ntdTomato:T NOS* cytokinin reporter assay, iP-NBD did not increase expression of the reporter 6 hours after treatment, but when applied simultaneously with iP, iP-NBD partially attenuated iP-mediated enhancement of the TCS reporter expression (Fig. S1d).

Thus, according to the pharmacological definitions (Jackson A., 2010, In: Stolerman I.P. (eds) Encyclopedia of Psychopharmacology. Springer, Berlin, Heidelberg), we believe that our various ways of testing point to the overall **partial agonist** character of iP-NBD. The molecule has weak agonistic activity not reaching effect of a full agonist and in the presence of the full agonist is competing for the same receptor and thereby reducing the full agonist effect; as presented in the lines 122-126 of the revised manuscript "*This suggests partial agonistic mode of action of iP-NBD that binds to a cytokinin receptor and activates it with only minimal efficacy compared to a natural cytokinin ligand. At excess concentrations, iP-NBD is then acting as a competitive antagonist, competing with the full agonist (a natural cytokinin) for receptor occupancy*".

We hope that with these additional analyses we can conclude that the iP-NBD exhibits activity corresponding to a partial agonist, although we would like to state our belief that in both cases,

whether agonist or antagonist the molecule binds the receptor, and thus it might serve as indicator of the receptor localization. Importantly, in our work to demonstrate the localization of the AHK receptors iP-NBD was used as a tool in combination with GFP-tagged receptor tracking and thus two independent methodologies were used to support our conclusions.

In ARR5 and TCS reporter assay, the incubation time is so long (16 h) to observe reporter protein expression. Authors also demonstrated that iP-NBD would be drastically metabolized after 1 hour (Fig. S2b). Thus, iP-NBD is continuously supplied into cells from a medium and iP-NBD would be decomposed during a long incubation to release small amount of iP, together with deprenylated Ade-NBD, and the iP released from iP-NBD might activate ARR5::GUS transgene.

Response: We understand the concern about long-term reporter assay experiments and risks related to metabolisation/metabolic interconversion of the iP-NBD and possible release of the iP.

Our measurements of the *in vivo* iP-NBD stability showed enzymatic deprenylation to Ade-NBD (Fig. S2b), but did not suggest a possibility that iP-NBD application contributes to the endogenous iP levels within the cells. Taking into consideration the reviewer concerns, we present an additional data from the iP-NBD turnover analysis that includes also measurements of iP, obtained from the same experiment. No significant change in the intracellular iP concentration was observed during the first 2 hours after iP-NBD application and, in opposite the iP endogenous level was significantly lower in the long-term view (please see detailed explanation and graph below, Revision Figure 2).

As described in the Supplementary methods, 5 μ M iP-NBD was applied to *Arabidopsis* (Ler) cell suspension and in the timeframe of 0.5 - 5 h (plus 20 h for long-term effect) samples were taken and deep-frozen for subsequent quantitative LC-MS/MS analysis of iP-NBD and iP contents. As shown in the graph below (Revision Figure 2) the intracellular concentration of iP-NBD and iP greatly differed from 2-4 thousand times. Whereas the intracellular iP-NBD level increased from app. 3,400 pmol/g (after 30 minutes of iP-NBD application to the cell medium) to app. 7900 pmol/g (within another 30 minutes), the intracellular concentration of iP was reaching in the same time points only about 2 pmol/g and, importantly, did not further change during the time. In opposite, with the prolonged time of iP-NBD application the intracellular concentration dramatically decreased (see the upper line of the Figure below). When we take into account that within 2 hours half of iP-NBD is metabolized (Fig. S2b), if release of iP is hypothesized, the endogenous levels of iP should greatly increase in this timeframe. As shown in the graphs below, this does not occur (see the bottom line of the Figure below).

Revision Figure 2: Intracellular concentrations of iP-NBD and iP in *Arabidopsis* (Ler) cell suspension after exogenous application of 5 μ M iP-NBD. Symbols refer to averages of four biological replicates, error bars represent SD. Upper row – comparison of the intracellular iP-NBD and iP levels during the whole time course of the treatment. Lower row – close-up to the period of time (first 2 h) when the cells contain highest amount of iP-NBD.

Furthermore, to avoid doubts about *in vitro* stability of iP-NBD and potential release of free iP during the treatment in the medium, we presented that iP-NBD is fully stable for at least 6 h in broad range of pH. In addition, during the revision we extended the analyses and tested iP-NBD stability in pH4 and pH6 (thus covering the range of pH expectable in the apoplast and in incubation medium) after 16 hours. We show that neither prolonged incubation significantly affect stability of iP-NBD (Fig S2a).

Because 10 nM iP is sufficiently induce ARR5::GUS same as 10uM iP-NBD (Fig. 1d) (0.1% of 10 μ M iP-NBD is equivalent to 10 nM iP). Although it is different systems, bacterial assay indicated that iP-NBD affinity to AHK4 is about 1000 times less active than iP (50% competition values in Fig. S1a) in competition assay, in contrast, 1uM iP-NBD can repress the TCS activation in root by 1 μ M iP (molar

ratio of iP : iP-NBD= 1:1) in Fig. S1c. The concentration of iP-NBD in this data (Fig. S1c) is correct? If it is correct concentration, how does authors explain these results.

Response: As shown in the manuscript, iP-NBD remains stable in the media (Fig. S2a) and intracellularly can be deprenylated to Ade-NBD by CKX (Fig S2c). Moreover, our analytical data presented above indicate that iP-NBD application does not contribute to the increase of endogenous levels of iP, thus they are not supporting assumption that some of the iP-NBD effects might be result of iP released from iP-NBD. It should also be taken into consideration that iP-NBD and iP not only compete for binding to the receptor, but also for entering to the cell (Fig. 1e and f). We can speculate that in some point the plants do not uptake more cytokinin if it is already saturated.

The concentrations used in Fig. S1d (molar ratio of iP : iP-NBD = 1:1) is correct. The explanation is connected to our very first response – the activity of iP-NBD can be result of partial agonistic activity with other receptor(s). The measured outcome of this assay is thus a result of a complex response of all the cytokinin-perceiving sites (receptors) differing in the affinity to iP-NBD ligand. Further, the real *ad hoc* molar ratio can be influenced by other factors connected not only to stability/metabolism of iP-NBD, but also to the changes in concentrations endogenous cytokinins after intensive feeding with iP into the plant. iP is metabolized too and might cause disbalance of the overall cytokinin status.

iP and iP-NBD are incorporated by same CK transport pathway (Fig. 1e) and iP-NBD is not stable for long time incubation.

Response: We cannot agree with this assumption. We show that iP and iP-NBD can be transported by (so far) the only well described cytokinin transporter PUP14. In general, we cannot exclude possibility that there are more ways how cytokinins and structurally similar compounds enter the cells. In our experiments, plants are kept in media as source of iP-NBD, where it remains stable for hours (Fig. S2a). We show that iP-NBD is only degraded intracellularly to Ade-NBD and therefore all the crucial experiments to monitor iP-NBD in cells were performed using only short-term exposures (up to 15-20 mins). Within this short time-frame plants are responding to the iP-NBD treatment by significantly increased expression of cytokinin primary response gene *ARR5* as shown in the newly added Fig. 1d.

Thus, iP-NBD would not be accumulated extremely higher than iP. However, iP-NBD can compete with iP at same concentration in root TCS reporter assay. These issues raise the question. iP-NBD really function on AHK in planta ?

Response: It is extremely challenging to monitor the concentration ratios of small molecules in spatio-temporal manner in living plant tissues and we are careful of making assumptions about the concentration ratios of iP and iP-NBD without experimental evidences. We believe that due to this technical limitations it is generally accepted to use simplification in the way we did in case of using reporter assays. We strongly believe that regardless this fact, we have done maximum to prove that iP-NBD functions on AHK in planta. We showed that a continuous provision of stable iP-NBD in the media is sufficient to provide a continuous source of iP-NBD to the plant that eventually can metabolize it at the intracellular level through enzymatic activity of CKX to Ade-NBD. We demonstrated by several approaches that iP-NBD binds the AHK receptor, and competes with natural cytokinin at several levels. Moreover, the newly added data of the short-term (within 15 minutes) response to iP-NBD at the level of expression of the primary cytokinin response gene *ARR5* (Fig. 1d) fully support the assumption that iP-NBD indeed functions on AHK *in planta*.

If iP-NBD has weak agonist activity in planta, the perception of iP-NBD by AHK occur at PM and ER. This case is simple. The iP-NBD is perceived at PM and ER that iP-NBD is accumulated. On the other hand, if iP-NBD is full antagonist (not agonist), the interpretation would be so complicated. Because

the experiment using antagonist (iP-NBD) always involve another molecule, agonist (iP). The transport, metabolic stability, endogenous/exogenous agonist, receptor selectivity, all these issues would affect the inhibition of agonist activity by the antagonist. I certainly know that this would be derived from technical limitations using small-molecule probes. To address the stability of iP-NBD during the reporter assay and mode of action of iP-NBD in planta, at least, I think the short-term response of plant to iP-NBD is essential data, instead of imaging analysis previously mentioned. Fig. S5e demonstrated that qRT-PCR results of ARR5 gene induction (15 min incubation). In the same way, the agonistic and antagonistic effects of iP-NBD in WT and ahk mutant should be evaluated at the same time-scale (15-40 min, stable condition for iP-NBD) as that used in the fluorescent imaging. It would convincingly support the binding of iP-NBD on AHKs in planta.

Response: As mentioned above, we followed the reviewer's suggestion and analyzed the effect of iP-NBD on cytokinin response in short time. We found that expression of the early cytokinin-response gene *ARR5* was significantly upregulated already 15 mins after exposition of Arabidopsis seedlings to several concentrations iP-NBD. As shown in the Fig. 1d, *ARR5* expression was increased by iP-NBD in the concentration dependent manner, although with app. 1,000-times lower efficiency compared to iP, in accordance with the observed activation of cytokinin reporter *ARR5::GUS* shown in Fig. 1d and the affinity of cytokinin receptors to iP-NBD (Fig S1a). Hence, we believe that altogether these results support the idea that that iP-NBD indeed interacts with cytokinin receptors. Simultaneous application of iP and iP-NBD did result in additive effect on the *ARR5* expression hinting at partial agonist activity of iP-NBD. This is also supported by the *pTCSn::ntdTomato:T NOS* cytokinin reporter assay. Although we could not detect iP-NBD enhancing effects on the expression cytokinin reporter (presumably due to lower sensitivity of this assay) 6 hours after treatment, when applied simultaneously with iP, iP-NBD partially attenuated iP-mediated enhancement of the TCS reporter expression (Fig. S1d). Thus as stated above, based on these results we conclude that *iP-NBD interacts with cytokinin receptors as a partial agonist*.

REVIEWERS' COMMENTS:

Reviewer #5 (Remarks to the Author):

In the revised version of the manuscript, the authors have successfully addressed all my concerns. The cytokinin activity of iP-NBD in planta is sufficiently demonstrated.